# Archaeological and molecular evidence for ancient chickens in Central Asia

Carli Peters [1], Kristine K. Richter[2], Shevan Wilkin [1,3], Sören Stark[4], Basira Mir-Makhamad[1,5], Ricardo Fernandes[1,6,7,8], Farhod Maksudov[9], Sirojidin Mirzaakhmedov[10], Husniddin Rahmonov[10], Stefanie Schirmer[1,5], Kseniia Ashastina[1,5], Alisher Begmatov[11,12,13], Michael Frachetti [14,15], Sharof Kurbanov[16], Michael Shenkar[17,18], Taylor Hermes [19,20], Fiona Kidd[21], Andrey Omelchenko[22], Barbara Huber [1,23,24], Nicole Boivin [1,25,26], Shujing Wang[27], Pavel Lurje [22], Madelynn von Baeyer [1,5], Rita Dal Martello[1,5] & Robert N. Spengler III [1,5] ✉

The origins and dispersal of the chicken across the ancient world remains one of the most enigmatic questions regarding Eurasian domesticated animals. The lack of agreement concerning timing and centers of origin is due to issues with morphological identifications, a lack of direct dating, and poor preservation of thin, brittle bird bones. Here we show that chickens were widely raised across southern Central Asia from the fourth century BC through medieval periods, likely dispersing along the ancient Silk Road. We present archaeological and molecular evidence for the raising of chickens for egg production, based on material from 12 different archaeological sites spanning a millennium and a half. These eggshells were recovered in high abundance at all of these sites, suggesting that chickens may have been an important part of the overall diet and that chickens may have lost seasonal egg-laying

Debate over the origin(s) and spread of domesticated chickens (*Gallus gallus* spp. *domesticus*) has intensified in recent years with the introduction of genetic and molecular methods, reigniting old controversies over the enigmatic bird[1–3]. Historical sources attest to the prominence of chickens in southern Europe and southwest Asia by the last centuries BC[4]. Likewise, art historical depictions of chickens and anthropomorphic rooster-human chimeras are reoccurring motifs in Central Asian prehistoric and historic traditions[5–7]. However, when this ritually and economically significant bird spread along the trans-Eurasian exchange routes has remained a mystery. Specialists agree that domestication traits evolved in an insular population of South Asian jungle fowl, likely the red jungle fowl (*G. gallus* ssp. *spadiceus*; involving hybrids of subspecies) somewhere across its expansive range from Thailand to India. However, scholars have also presented widely diverging dates and routes of spread, and some of this confusion comes from unclear identifications of birds in ancient art historical depictions[3] and overlap in morphological features of chicken bones

with those of certain wild avian species. In addition, brittle hollow bones and eggshells are far less likely to be preserved, recovered, and identified than those of other animals[8]. Further blurring the narrative of the early stages of chicken spread is the fact that pheasants (Phasianidae) and ducks (Anatidae) in China and geese (*Alopochen aegyptiaca*) in Egypt have, at times in the ancient past, been maintained (but not necessarily phenotypically altered) or heavily hunted. These chicken-sized birds can easily be misidentified in zooarchaeological studies. Biomolecular techniques, such as peptide mass fingerprinting and ancient DNA are uniquely poised to overcome these taphonomic and morphological obstacles and to clarify one of the remaining mysteries in the domestication and spread of animals across Eurasia.

Many scholars have argued that chickens occupied a symbolic and social domain prior to the Hellenistic period (fourth to second centuries BC), after which point their bones sharply rise in ubiquity and abundance in archaeological assemblages[1]. One often-propagated claim is that the chicken served an entertainment role in cock fights

before it took on its economic status, and some scholars have hypothesized that breeding sports birds may have segued into their role as food[9,10]. Historians have also suggested that chickens were a sacrificial animal for divination or an elite commodity for their plumage and as live caged displays prior to their dispersal across West Asia[11]. They maintained a ritual role into the Roman period as attested from Classical texts and archaeological contexts.

Zooarchaeologists claim to have identified domesticated chickens at a handful of southwest Asian sites from contexts dating to the late second and early first millennia BC[2,12]. However, these finds consist of one or two bones in assemblages of tens of thousands of animal bones (meriting reevaluations). Recent radiocarbon dating and new assessments of identifications have consistently illustrated errors in the early dates[13,14]. The earliest sites with well-identified chicken bones that appear in abundance are Tel Kedesh (last centuries BC[2]) and Maresha (4th–2nd centuries BC[1]), both in Israel (Fig. 1). From contexts dating only a few centuries later, the bird is reported from archaeological sites in Japan and Korea—the Yayoi period (ca. 300BC–AD300[14]) and in England by the first centuries AD[15]. Some historians have suggested that the chicken spread through the Mediterranean with Phoenician traders during the mid-first millennium BC. Other scholars suggest that the expansion of the Persian Empire by Cyrus in 539 BC or the slightly later Macedonian expansion (330s BC) opened the long-distance exchange networks that allowed the chicken to cross the world[8]. Debate of early chicken spread focuses on the reliability of reports of chicken bones in pre-Hellenistic contexts. For example, there have been many claims of pre-Ptolemaic chickens in Egypt, but all evidence older than the last centuries BC has been either questioned or rejected by scholars[16]. Fascinatingly, the chicken appears to have spread via coastal routes to Central Africa prior to its dispersal in Egypt[17]. There are also a handful of early reports of chicken bones in Europe, pre-dating the Roman expansions, but only rising to prominence in the Greco-Roman period, when specialized poultry farms developed for the first time[18,19]. Chicken bones are a common feature at nearly all Roman sites, including ritual centers, villages, and farmsteads, and the Classical importance of chickens is well attested in textual sources[20].

In this work, we illustrate how a combination of historical, archaeological, morphological, and palaeoproteomic analysis can aid in identifying ancient chicken eggshell fragments, which we have compiled from twelve archaeological sites in Central Asia, spanning ca. 400 BC to AD 1220. The lack of eggshells from any older Central Asian archaeological sites (for an updated list of sites where flotation and water screening work has previously been conducted in Central Asia see refs. 21–24) hints to a rapid rise of egg-laying and chicken rearing across Iranian West Asia (Hellenistic and Zoroastrian traditions), an economic practice that appears to have remained prominent through the medieval period in this part of the world. The sites in our study have all been well-dated with extensive radiocarbon sequences complementing ceramic and numismatic seriations.

Here, we show that the cultural shift in the role of the chicken to an important food source across the ancient world was tied into the development of more prolific egg-laying phenotypes combined with the expansion of the Persian, Macedonian, and Roman Empires, along with their associated trade networks. We present new evidence for a prominence of chicken egg production in Central Asia starting in the last centuries BC and continuing into the medieval period. The archaeological data that we present consist of fragments (1.5–4.0 mm) of eggshells (Fig. 2) recovered from sediments in archaeological contexts at the core of the ancient Silk Road trade routes. The data come from Bash Tepa, a late Achaemenid through Hellenistic fortified site on the edge of the Bukhara Oasis (ca. 400

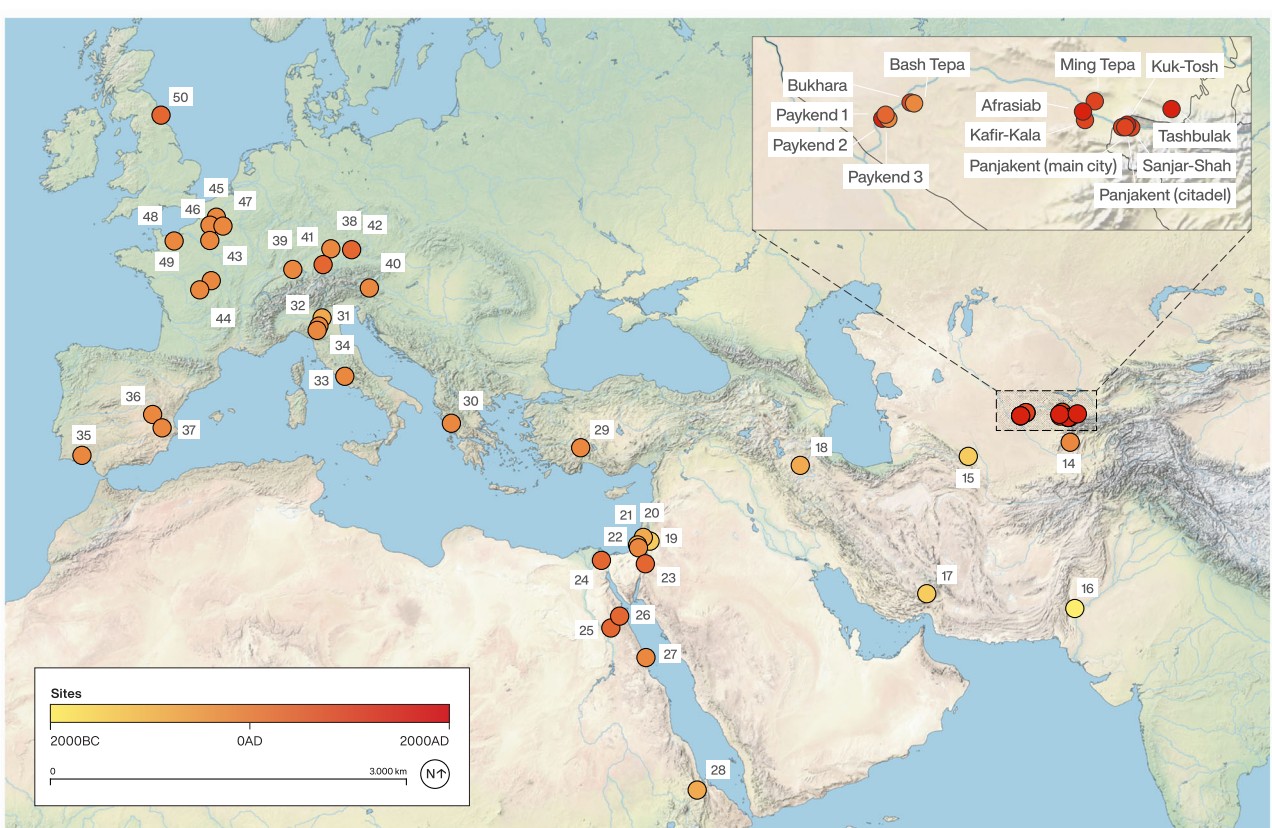

**Fig. 1 | Key data points in the dispersal of chicken.** Information for this map is provided in Supplementary Data 1. Given the quantity of evidence for chickens in Roman and medieval Europe and West Asia, this map is not comprehensive of early finds of chickens, but rather provides an idea of the rapid rate of adoption of chicken rearing.

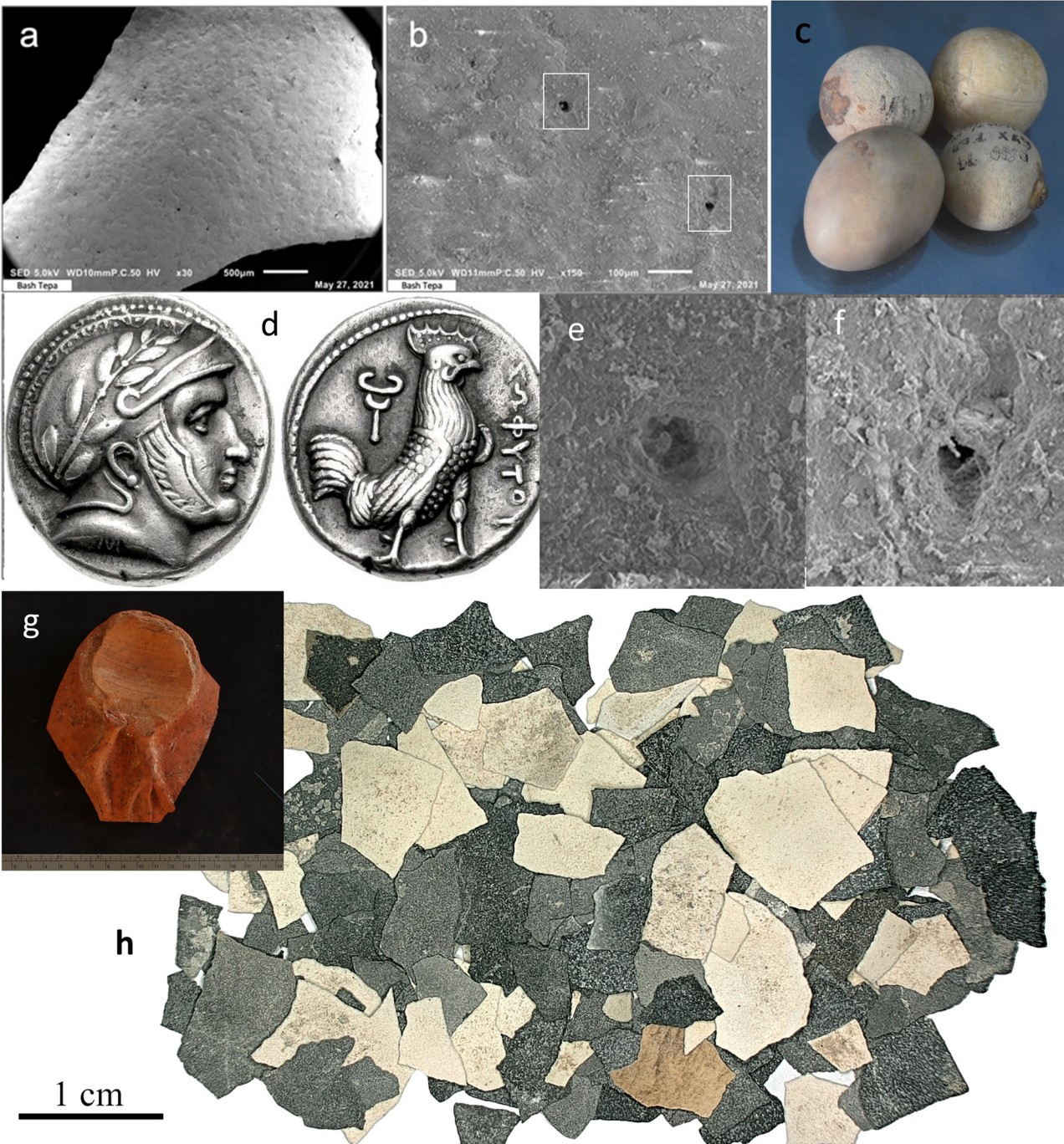

**Fig. 2 | A compilation of evidence for ancient chickens in Central Asia.** Including: SEM images of a Bash Tepa egg shell, emphasizing morphologically distinct breathing pores at magnifications ×30 (**a**), ×150 (**b**), ×750 (**e**, **f**). **c** A ceramic egg with clay balls from Bukhara dating between the tenth and twelfth centuries AD; **d** the Sophytes coin from Bactria in 300 BC (This image is licensed under the Creative Commons Attribution-Share Alike 3.0 Unported license: https://commons. wikimedia.org/wiki/File:Sophytes_hemidrachm.jpg Attribution: Classical Numismatic Group, Inc. http://www.cngcoins.com); **g** a fragment of an ossuary from Bash Tepa dating to the last centuries BC, with an apparent chicken on the top; whole examples of similar funerary urns have been identified at other sites in Central Asia; and (**h**) a selection of eggshells from the Bukhara site, showing color (essentially all white) and burning, which was evident on many of the shells.

BC–AD 100), several occupation layers at Paykend dated from the Hellenistic (ca. 4th–2nd centuries BC) to medieval period (10th–12th centuries AD), the medieval cities of Kafir Kala (4th–12th centuries AD), the inner city (shahristan) of Bukhara (10th–11th centuries AD), Afrasiab (10th–12th centuries AD), ancient Panjikent (citadel and main town, 5th–8th centuries AD), Kok-Tosh or pre-Mongol Panjakent (9th–12th centuries AD), Sanjar-Shah settlement (5th–9th centuries AD) and the high-elevation urban site of Tashbulak (10th–12th centuries AD). A sampling of eggshells were taxonomically identified with peptide mass fingerprinting. Eggshell proteins can preserve over long periods[25] and are highly variable between taxa[26], and can thus be used to taxonomically identify archaeological eggshell remains. Given the dearth of previous research on this topic in Central Asia, this study fills an important and rather large gap in knowledge. We propose two hypotheses: (1) the rapid and widespread dispersal of chickens across the ancient world during the last centuries BC coincided with greater egg-producing variants; and (2) poultry farming and egg production were an important and regular part of the economy in villages and

urban sites across Central Asia from the Hellenistic through at least the Qarakhanid periods.

## Results

### Archaeological eggshells

Eggshells were recovered from all archaeological sites that we examined in this study (Fig. 1), all of which are located along the core exchange corridor of Central Asia dating from the last centuries BC through the first millennium AD. In addition, eggshell fragments are absent from any earlier occupation sites in Central Asia that have thus far been examined using sediment flotation (considering that absence of evidence is not necessarily evidence for absence). The eggshell fragments were recovered from anthropogenic sediments using a 1.4 mm geological sieve as part of the heavy fraction recovery portion of archaeobotanical studies. A summary of the recovered remains and the density of shell fragments in the sediments are presented in Table 1 (see Supplementary Data 2 for a more detailed overview). All eggshells were white in color, except for a few from Tashbulak that appeared to exhibit a speckling coloration, which may represent either a distinct variety of chicken or post-depositional staining (Fig. 3). The data used in this study come from 13 archaeological sites, broadly speaking, 12 of which are urban or village sites and have been associated with agropastoral communities. One additional site, LVD-HA-K7 (in the Bukhara Oasis), is a burial mound and was linked to a different population of people than those living in the urban sites. Two archaeological sites date to the Hellenistic period, Bash Tepa and Paykend 1; both of which had eggshells in their sediments in relatively high densities (Table 1). In Samanid and Qarakhanid period sediments, we recovered eggshells at significantly higher densities, being prominent in samples from Kafir Kala, Bukhara, Afrasiab, Ming Tepa, Tashbulak, Panjakent 2, and Paykend 2 and 3. To ensure that the oldest dates for our recovered eggshells are of the same age as the dated archaeological deposits, we ran two direct radiocarbon dates on the oldest material from two of the sites, resulting in calibrated dates of cal. 515-392 BC (95.4%; Paykend [2370 +/− 20]) and cal. 385-206 BC (95.4%; Bashtepa [2240 +/− 20]). Collectively, these data suggest continual and regular deposition of eggshells across all sites and throughout the sediment accumulation period.

### Proteomic identification of eggshell

Peptide mass fingerprints were obtained for eggshell fragments from five archaeological sites, Afrasiab ($n = 5$), Bash Tepa ($n = 5$), Paykend ($n = 5$), Tashbulak ($n = 1$), and burial mound LVD-HA-K7 ($n = 2$) (see Supplementary Table 1 for a detailed overview of the sample numbers, context information and identification). The majority of the samples (16 out of 18) were identified as chicken based upon published peptide markers (Fig. 4). The identification to *G. gallus* was confirmed with LC-MS/MS analysis of one of the samples to rule out locally present galliform species for which published MALDI markers do not exist. A reference database including sequence data for all common eggshell proteins for all bird species, and all available Galliform proteomes, was used to rule out closely related members of the Phasianinae subfamily (Phasianini, Tetraonini, and Coturnicini) that are locally present in Central Asia (Supplementary Table 2) as possible identifications of the eggshell fragments. Of the top 30 recovered proteins that derive from eggshell, after excluding those from soil or laboratory contamination or those that were not unique, three belong to the infraclass of Neognathae (majority of living birds), five belong to the Galliformes order (heavy bodied, ground-feeding birds), six belong to the Phasianidae family (ground living birds), and ten are specific to *Gallus gallus*. For all protein IDs, peptide spectral matches, peptide sequences, and taxonomic identifications see Supplementary Data 3. See Supplementary Methods for an in-depth discussion of the proteomic identification, and Supplementary Data 4 and 5 for a detailed overview of the peptides and proteins identified in the sample and their uniqueness.

All of the chicken samples were from residential contexts. The only two samples not identified as chicken were from a non-residential context, burial mound LVD-HA-K7. One sample was identified as Anseriformes, an order of waterfowl birds including ducks, geese and swans, based on earlier published markers[26]. The other has a low-quality spectrum more consistent with Anseriformes than chicken, but there were too few peaks present for confident taxonomic identification. Within Anseriformes, it is not possible to identify either sample to a more precise taxonomic level, since the majority of the MALDI peptide markers published are identical in all studied species from this order[27].

**Table 1 | Eggshell densities (number of eggshells per liter of sediment) and ubiquities (number of sediment samples with eggshells per site) by site and date**

| Site | Number of screeneds amples[a] | Age of samples | Liters of sediment | Number of eggshell fragments | Ubiquity | Density |
|---|---|---|---|---|---|---|
| Bash Tepa | 30 | 400 BC– AD 100 | 764 | 460 | 0.7 (70%) | 0.602 |
| Paykend 1[b] (Citadel) | 10 | 5th century BC–5th century AD | 350.5 | 20 | 0.3 (30%) | 0.057 |
| LVD-HA-K7 (Burial Mound) | Handpicked | 1st century BC–1st century AD | | | | |
| Panjakent (Kainar-citadel) | 9 | 5–7th centuries AD | 116.5 | 25 | 0.88 (88%) | 0.215 |
| Kafir Kala | Handpicked | 4–12th centuries AD | | | | |
| Panjiakent (Main town) | 11 | 7–8th centuries AD | 264.5 | 56 | 0.64 (64%) | 0.211 |
| Ming Tepa | Handpicked | 7–8th centuries AD | | | | |
| Sanjar-Shah | 5 | 8–9th centuries AD | 65.5 | 26 | 0.8 (80%) | 0.397 |
| Kuk-Tosh | 6 | 9–12th centuries AD | 79 | 2,847 | 0.83 (83%) | 36.038 |
| Bukhara | 26 | 9–12th centuries AD | 939.5 | 871 | 0.46 (46%) | 0.927 |
| Tashbulak | 22 | 10–12th centuries AD | 223.5 | 95 | 0.23 (23%) | 0.425 |
| Afrasiab | 1 | 10–12th centuries AD | 255 | 855 | 1 (100%) | 3.352 |
| Paykend 2 (Shakhristan) | 2 | 10–12th centuries AD | 55 | 101 | 0.5 (50%) | 1.836 |
| Paykend 3 – (Rabat4) | 20 | 10–12th centuries AD | 357.5 | 90 | 0.5 (50%) | 0.251 |

[a]Paykend 1 and 2 represent the Citadel and Shakhristan II areas excavated during 2019; while Paykend 3 represents the neighboring medieval caravansary (Rabat-4). Samples from Kafir Kala and LVD-HA-K7 were handpicked and therefore ubiquities and densities cannot be calculated.

[b]In this case, a sample refers to a large sampling of sediments, in many cases close to 10 liters, from anthropogenic contexts.

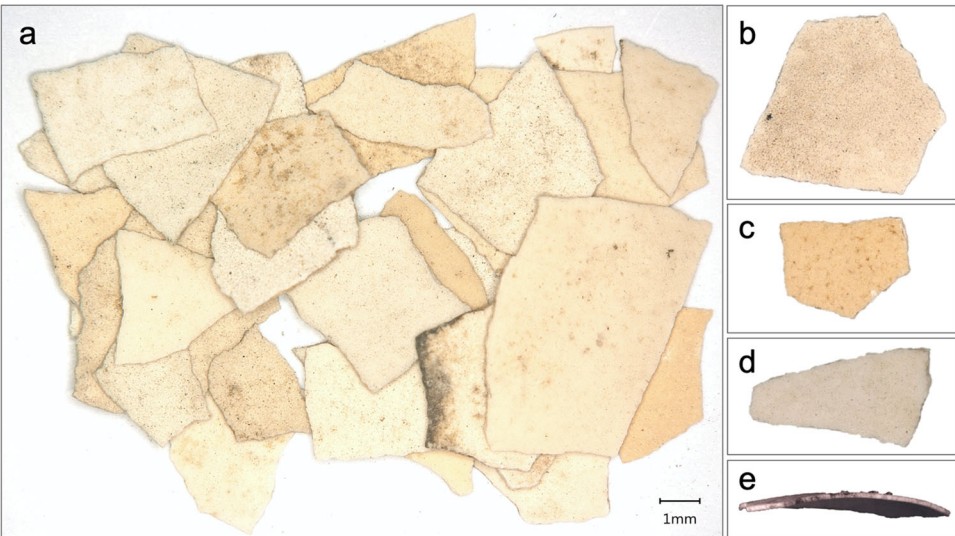

**Fig. 3 | Image of the fragments of ancient eggshells, showing the shape and surface coloration as well as the general curvature of the fragments. a** Eggshells from the Bash Tepa FS2 sample; **b** an eggshell from the Paykend FS8 sample (midden); **c** an eggshell from the Baykend FS15 sample (rabat); **d** an Afrasiab eggshell; and (**e**) a Tashbulak eggshell.

## Discussion

### Potential evidence for non-seasonal egg laying

We present evidence for ancient chicken eggshells from 13 different archaeological sites, spanning a period of a millennium and a half (Table 1). All analyzed eggshells were identified as chicken, except two fragments recovered from the same context in a burial mound (LVD-HA-K7), which have conservatively only been identified as Anseriformes, presumably a wild waterfowl. This does not necessarily rule out the possibility that other birds, such as geese were raised at these sites. Follow-up zooarchaeological research will hopefully further clarify the economy of poultry raising in ancient Central Asia. More informatively, eggshells were recovered at high ubiquity from these sites, meaning they were located in most of the stratigraphic layers and archaeological contexts that we examined. It is well-accepted by archaeobotanists that artefacts recovered in many different archaeological contexts are more likely to represent frequent-occurrence events, as opposed to seasonal or semi-annual events[28–31]. In archaeobotany, this reasoning is often applied to grain chaff, suggesting that high ubiquity of remains likely represents a situation where a household processes grain daily and stores the crop in its chaff, as opposed to one seasonal grain threshing event and the storage of grain in a cleaned state. The same reasoning could apply to the eggshells in these assemblages, indicating that they were deposited more frequently than during a single seasonal laying event. While further research is needed to either verify or refute this hypothesis, the high abundance and ubiquity, may illustrate the importance of eggs in the dietary economy, as a common food over a greater period of the season than would be expected for behaviorally wild fowl. The wild reproductive cycle of a chicken progenitor consists of one brood of eggs a year, with a clutch of less than 6 eggs[8]. The resulting evidence may suggest that these chickens expressed shifts in reproduction from the wild and could have been producing eggs at a regular rate for a significant part of the year; although, these data do not allow us to specify the duration or abundance of laying.

While few archaeological projects have specifically sought out eggshell fragments in anthropogenic sediments, they are occasionally reported. However, evidence from before the second century BC for chicken eggs is completely lacking. Eggshell fragments were reported from Roman Mons Claudianus in Egypt, and Van Neer et al.[32] argued that the prominence of medullary bones at the Roman site of Berenike is indicative of an egg-focused chicken industry. Evidence for egg production in Roman Britain comes from tablets found at Hadrian's Wall, noting receipt of, among other items one or two hundred eggs (chickens are among the other listed items[33]). More informatively, eggshells have been identified at 38 Roman period sites in Britain[34]; using a combination of SEM and peptide mass fingerprinting identification techniques, the eggshells from the amphitheater at Chester, Cheshire (AD 70–80), were identified as chicken[33]. Further evidence for egg production in the Roman period comes from Diodorus Siculus, who mentioned a way to incubate chicken eggs. Columella[19] (Book VIII: II:3–8) discussed specialized chicken farmers and suggested that certain kinds of chickens were better for cockfighting and others for egg laying. Some of the breeds he mentions appear to have originated on islands, and may have already diverged through insularity over the previous few centuries. He specifically references a form of poultry from Adria, near modern-day Venice; the Adrian chicken is also noted by Pliny the Elder[10]. Aristotle's[35] famous chicken embryo studies may further attest to the prominence of eggs in ancient Greece. Eggs are mentioned in Apicius' cook book[36] (Book 6, chapter 248:2–3). Pliny claimed that the best birds could lay daily, but most historians agree that this is a significant exaggeration, as were many of Pliny's claims[20]. In addition, Varro[37] (Book III, 481) discusses how to care for hens when they are laying and further illustrated the prominence of chicken egg production by the early Roman period.

A recent aDNA study explored the increase in prominence of a derived variant in TSHR (thyroid-stimulating hormone receptor) among archaeological chicken remains from Europe. They suggest that this allele shift would have been associated with reduced aggression and faster onset of egg laying[38]. They further argue that these traits would only have been selected for beginning around 1100 years ago, emphasizing that this means many of the traits associated with modern domesticated chickens are recent in origin. They note that TSHR, in part, regulates thyroid hormones, which in turn alter growth, metabolic regulation, and photoperiod control. The change in metabolic regulation, notably associated with reproduction, has long been accepted as one of the key variables of domestication in the chicken[39]. As a final line of evidence for a deeper legacy of egg-laying chickens in West Asia, historical landraces or ecotypes attest to a long legacy (stretching back at least for several centuries) of egg-laying chickens. Genetic studies suggest that there has been long-term isolation and maintenance of insular breeds in remote areas of Iran and Turkey[40,41]. This deep continuity of egg-laying breeds seems to support

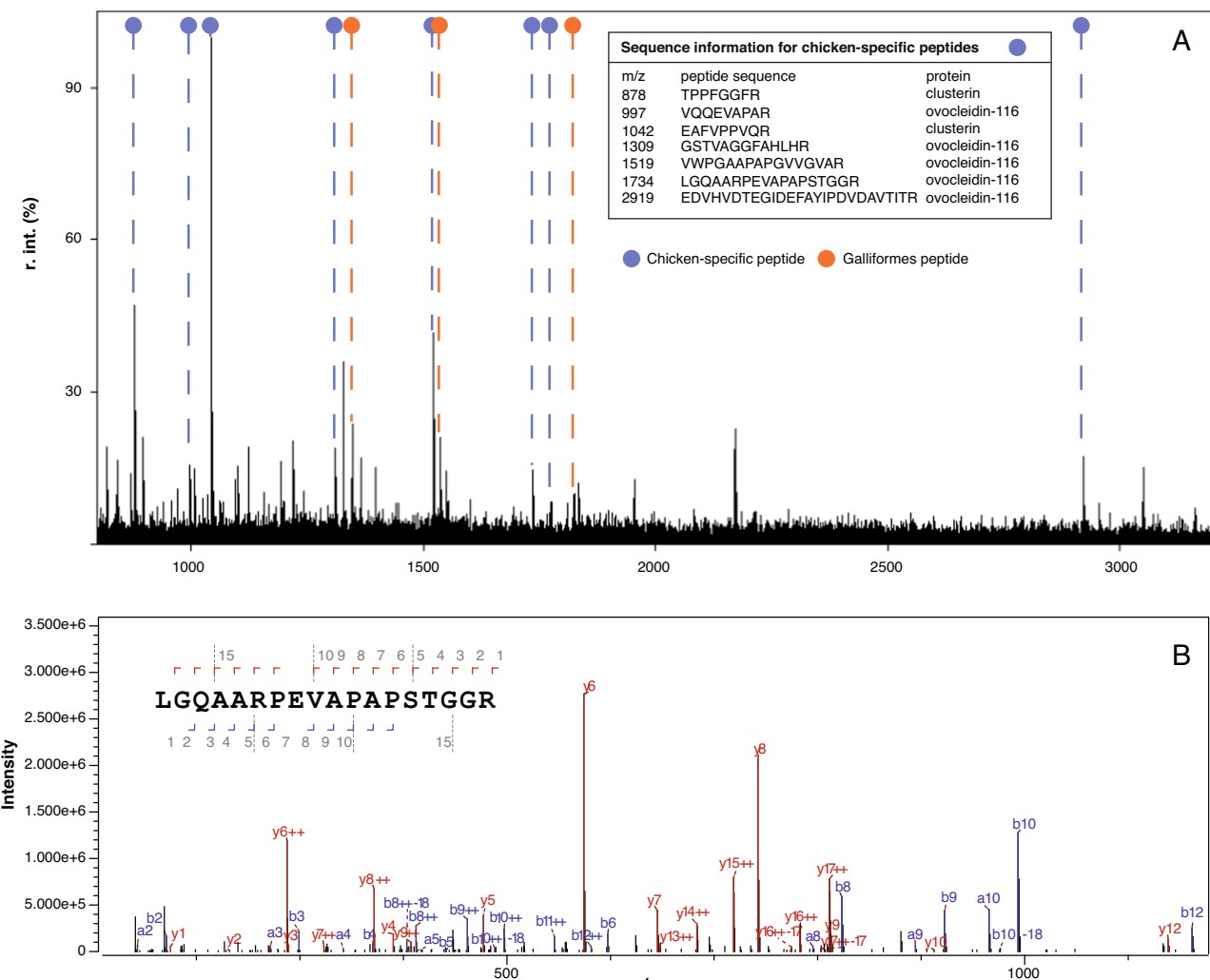

**Fig. 4 | Results of palaeoproteomics analysis. A** MALDI-ToF-MS spectra of sample CP565 from Afrasiab. The chicken-specific and Galliform peptides identified in the sample are highlighted. Sequence information was taken from Presslee et al.[26]. **B** Example of MS/MS data with the b and y ion series for one of the chicken-specific peptides identified in the sample.

our argument that the traits of multi-annual reproductive cycles evolved before the dispersal of chickens into Europe. That said, the rapid and ongoing extinction of historical landraces in southwest Asia is evidence that recently developed (within the last century) breeds are far more productive than the ancient ones[40]. In fact, modern commercial poultry production has largely erased the genetic legacy of ancient chicken landraces or ecotypes globally, further complicating studies of the origins and dispersal of this enigmatic animal[42].

### Conflicting Claims for Chicken Origins

The domestication of the chicken remains a largely unresolved topic; zooarchaeologists and geneticists have proposed many highly divergent narratives. Origins have been presented as being in Burma[43], India[12,16,44], Thailand[45], and northern China[46], as well as in southeast Asia more than nine millennia ago, with an early dispersal into northern China by 6000 BC[12]. Among the leading theories, introgressive hybridization has been invoked, suggesting gene transfer from *G. gallus* ssp. *sonneratii* to the modern chicken lineage[47]. Alternatively, a hybrid complex of multiple subspecies, excluding *G. gallus* spp. *varius*, has been proposed[48]. Other geneticists have proposed a hybrid complex of different subspecies originating from differing regions, "such as Yunnan, South and Southwest China and/or surrounding areas (i.e., Vietnam, Burma, and Thailand) and the Indian subcontinent"[49]. More

recent genetic work has suggested complex hybrid origins from the crossing of at least three wild lineages, with separate domestication events across southeast Asia and India[50], while others have suggested that the main genetic contribution for the modern chicken came from *G. gallus* ssp. *gallus*, with continual bidirectional gene flow between that lineage and *G. gallus* spp. *lafayettii* and a single introgression event from *G. gallus* ssp. *Varius*[51]. Another prominent set of genetic models postulates that all modern chickens are monophyletic[45,52]. Other claims place the origin of domestication in "southeast Asia nearly 10,000 years ago"[53]; another postulates a 5,400-year-old origin in southeast Asia, with multiple isolated lineages following distinct routes of dispersal[42]. A mtDNA study concluded that they had identified "several local domestication events in South Asia, Southwest China and Southeast Asia", and that their genetic data illustrated how the chicken helped early peoples colonize the Pacific[54] (227). Another recent approach stated "Several domestication centres have been identified in South and South-East Asia. *Gallus gallus* is the major ancestor species, but *Gallus sonneratii* has also contributed to the genetic make-up of the domestic chicken"[55] (197).

Further disagreement has occurred over how the initial steps towards domestication unfolded, with one team of geneticists pushing for "intensive breeding and selection programmes"[56] (285), whereas many other scholars have suggested that the first steps involved

unconscious hybridization or commensalism. Some geneticists have argued for intentional and conscious selection of higher egg-yielding birds in prehistory[50,57]. Other scholars evoke a complex interplay between a protracted process, continual founder effects, and a more recent selection for greater production[48,57]. As noted, many historians and archaeologists have suggested that the chicken was first domesticated for sport or for ritual and not for food[58]. Some of the most recent discussions on this topic suggest recent and truncated domestication processes, with the first steps in the process occurring over a period of less than a millennium and the earliest dispersal of chickens involving ornamental exotic forms, as opposed to rapid growing and high-egg-yielding forms[13,14]. The remains from Mohenjo-Daro have been widely referenced as conclusive evidence for domesticated chickens in the Indus by 2000 BC[12]. However, the discovery consists of only a few bird bones, recovered from the upper levels of the site roughly a century ago[59]. This often-cited data point was recently reassessed by Peters et al.[14], who claim that the early bones are morphologically not from chickens. As with many domesticated organisms, the molecular clock results are not overly reliable, providing a range for the earliest divergence of the domesticated chicken from its wild lineage of ca. 9500 ± 3300 years ago[3]. The most recently published modern genetic study, applying a global genomics approach, concluded that the red jungle fowl (*Gallus gallus* ssp. *spadiceus*) is the main progenitor[14]. A recent redating campaign by Best et al.[13] and Peters et al.[14] from roughly 600 sites, spanning 89 countries, is now pushing for a recent domestication, specifically in central Thailand, occurring roughly between 1650 and 1250 BC with the earliest possible spread into Europe of 800 BC.

Over the past few years there has been an ongoing published debate over a possible Chinese center of domestication. West and Zhou[12] argued for domesticated chicken zooarchaeological remains from the archaeological sites of Peiligang and Cishan, dating to roughly 6000 BC. In 2014, a team of scholars claimed to recover mtDNA for domesticated chickens from the site of Nanzhuangtou, dating back more than 10,000 years and boasting a new origin of domestication on the central plains of China[46]. Xiang et al.[46] further supported these claims and providing additional genetic evidence for domesticated chickens at Cishan and Peiligang, dating to more than 7000 years ago.

Peng et al.[60] were the first to question the claims, disputing the ability to verify that the bones were from a domesticated species, given the short mtDNA sequence, Xiang et al.[61] replied in agreement that their genetic methods were inadequate; however, they still defended their overall conclusion by stating: "Even if the mtDNA sequences from Nanzhuangtou and Cishan are from wild junglefowl populations, they support the conclusion that chicken domestication would have been possible in northern China at that time."[61] (E1973). Peters et al.[62] also questioned the claims for a Chinese center, using traditional zooarchaeological methods to suggest that the bones may be from pheasants despite the genetic evidence presented in the original paper. Further focusing on osteological morphology, a team of specialists pushed the idea that the early remains from Shenmingpu in Henan were from ring-necked pheasants (*Phasianus colchicus*; Deng et al.[63]). A year later, a much more detailed study, based on well-established morphological approaches, looked at 1831 bird bones, 429 of which were previously recorded as domesticated chickens; the bones came from 18 different Neolithic and early Bronze Age sites in central and northern China[64]. Peters et al.[65] also provided a critical review of the claims of early Chinese domestication and disputed the paleoclimatic assessments from the original debate, which had implied that the red jungle fowl may have existed as far north as the central China plains. Following up on this debate, a new mtDNA analysis of the earliest purported chickens from the Dadiwan site in northern China identified the species as pheasant (*P. colchicus*)[66].

A large-scale reassessment provided by Peters et al.[14] further discredited any published early dates for chickens in China, to which

Peng et al.[67] responded agreeing that the new earliest date for domesticated chickens come from Thailand at 1650–1250 BC, but noting that this still does not exclude China as the origin zone. They then proceed to suggest that bones recovered from the site of Dadianzi, astonishingly rather far north, in Inner Mongolia could represent the early chickens. They also suggest that two early sites in Yunnan need to be considered, Caiyuanzi and Dadunzi, both dating to slightly more than 4000 years ago. However, they also present an interesting photo of a bronze rooster that is dated, based on style, to around the Shang period, older than 3,000 years ago and recovered from Sichuan. Peters et al.[68] replied, by simply noting that they read the original site reports for the three sites that Peng et al. mention, do not provide direct dates, and the single bone from Dadianzi is described in the original publication as "possibly domesticated". Ultimately, the bronze figurine may turn out to be the oldest evidence for chickens or possibly jungle fowl in southern China.

## Dissemination across the Ancient World

As with the earliest evidence for domestication, there are many claims of early discoveries of domesticated chickens in various regions of Eurasia. In most cases, these early dates cannot be refuted without further investigation. As one example, Meadow[69] suggested that chicken bones were present at Tepe Yahya in Iran dating to 3900–3800 BC, but noted that they are only prominent in the assemblage after 1000 BC Peters et al.[14] recently rejected the early Tepe Yahya chickens. Likewise, a bundle of bones recovered from an eighteenth Dynasty (1320 BC) tomb in Egypt, were reassessed from being chicken bones to belonging to a mix of waterfowl and predatory birds[70]. Other scholars have directly dated chicken bones and found that they were intrusive from later occupation layers[13,14,71,72]. Best et al.[13] recently ran 23 radiocarbon dates on early chicken remains from 16 sites, discovering that 18 of them were erroneously dated. As part of this bigger reassessment, Peters et al.[14] further rejected most of the early dates across Europe, Asia, and North Africa—suggesting that dozens of published finds of early chickens are wrong. Pre-first millennium BC claims of chickens in West Asia and Europe always appear in very low abundances, for example, at the Hesban site, in Jordan, a few bones have been reported to be chicken and dated between 1200–900 BC, and the earliest reported chicken from Israel consists of a single bone from Shiloh dated to 1650–1550 BC[0]. Out of nearly 5500 identified bones from Tel Michal, only one chicken bone was reported[73]; from Tel Lachish, out of more than 27,000 identified bones, many of which were from birds, two were identified as chicken, both coming from late occupation layers[74]. In sharp contrast, at the Hellenistic site of Tel Kedesh in northern Israel, in the last centuries BC, 310 chicken bones were recovered and Redding[0] argued, based on the prominence of cortical bones, that they are linked to more prominent egg laying.

While the chicken appears to have been known and traded in the eastern Mediterranean at least as far back as the eighth century BC[2,58], the earliest indisputable evidence for the bird taking on a significant role in the dietary economy comes from the site of Maresha, Israel, a Hellenistic village dating between the fourth and second centuries BC[1]. The Paykend and Bash-Tepa eggshells presented in this paper are contemporaneous with the bones from Maresha. Seemingly, the rearing of chickens for eggs rapidly spread across the eastern Mediterranean during and shortly after the fourth century BC. Chicken bones have been recorded at archaeological sites around the Mediterranean from as early as the eighth century BC, notably at Phoenician sites in Iberia[72,75,76]. Detailed and indisputable images of roosters, often in fighting scenes, are present on Greek pottery, notably Corinthian wears, dating back to 620 BC[8]. The bird was clearly present across Greece, France, and southern Europe through the second half of the first millennium BC; although, it remains a rare occurrence in these sites until the last centuries BC[77,78]. Additionally, Theognis of Megara

mentions the rooster around the sixth century BC[79]. It seems likely that these rare finds of pre-Hellenistic chickens were raised for meat, sport, ritual purposes, or as exotic prestige items, as opposed to egg laying. For example, at a second century AD shrine to Mithras in Germany than 7500 chicken remains were recovered[80] (the linkage between the chicken and a deity associated with Persian/Iranian origins is unlikely to be a coincidence). Cicero mentioned their power of divination in De Divinatione[81]. The imagery of cock fighting also continued into the Greco-Roman period and accounts for many of the earliest chicken images in the Mediterranean, such as on coins from the Temple of Artemis at Ephesus, estimated to date to 625–600 BC[10] or the elaborate mosaic depiction of the cockfight in House of the Labyrinth at Pompeii.

The map in Fig. 1 illustrates the dissemination process, but it does not present an exhaustive survey of archaeological chicken bones recovered from Roman sites, because chickens were clearly a fixture at all villages, cities, and small homesteads across the Empire. The most cited historical reference to the timing and routes of chicken dispersal comes from the Greek author, Athenaeus (third century AD), who claimed to be referencing an earlier source by Cratinus in referring to the bird as the Persian bird. Likewise, Aristophanes[82] in, *The Birds*, refers to the chicken as the Median bird. References to roosters are too numerous in Classical Roman texts to enumerate here. As a few select examples, there are references to chickens by Pliny the Elder, Virgil, Pliny the Younger[83], Varro[84], and Cicero[81], most of whom refer to cock fights. However, historians have emphasized the economic importance of chickens in the Roman Empire, suggesting that they were being cultivated on a large scale and that specialized poultry farms existed[18]. The most often cited supporting text for this claim comes from Columella (ca. AD 60), who described in detail the practices of poultry raising in the Greco-Roman world[19].

Many claims for early chickens in Egypt have been made based on images of birds in art or hieroglyphics[0], but these require further verification. Chickens are present in the faunal assemblage from Berenike during the Ptolemaic period, but they increase in abundance threefold during the Roman period[85]. One pre-Ptolemaic report of possible chicken bones in Egypt comes from the site of Buto (685–525 BC[86]). Clearer chicken evidence comes from Coptos in Greco-Roman sediments (150 BC–AD 300[87]). Robust evidence for chickens in Egypt during the Roman period comes from sites, such as Tell Maskhuta (AD 400[88]) and Quseir (first century BC to early sixth century AD[89]). Some archaeologists working in Egypt have argued that the chicken was introduced early, lost, and then reintroduced during the Ptolemaic period[16], others dismiss the fragmentary earlier evidence, claiming that the chicken was first introduced to Egypt with the Greeks[70]. There is, however, good evidence for chickens in Central Africa earlier; at the pre-Aksumite site of Mezber, dating between 800 and 400 BC, in Ethiopia in the Horn of Africa[17].

Zooarchaeological remains for chickens suggest that it spread into Central Europe by the Hallstatt C–D period ca. 800–475 BC[71]. One site that has provided strong evidence for pre-Roman chickens in Central Europe is Manching, a Celtic oppidum (Fig. 1), which has provided a massive assemblage of zooarchaeological remains, including rare finds of chicken bones dating to the La Tène period (475–30 BC[90]). Chicken remains are identified in very low frequencies, notably at southern village sites dating to the last two centuries BC[91]. Although, some scholars have argued that chickens were kept more for ritual offerings than as food in pre-Roman Britain[33]. The chicken has been identified at Roman ports as far afield as Arbeia, England, from between AD 150 – 450[15]. A large-scale zooarchaeological synthesis of chicken remains from the British Isles notes that they became slightly more prominent during the Roman period, but were most prominent in graves, shrines, and ritual deposits[33]. Chickens have been identified in other areas of the Roman Empire, including Italy[92] northern France[93], Switzerland[94], and northern Africa[95], and even far northern Europe[96]. A recent synthesis of the zooarchaeological literature for Russia has

illustrated that the chicken spread into western Russia and the river valleys of the steppe by 1,000 years ago and aDNA data suggested that it spread into the region from Europe rather than West Asia[97]. Despite the lack of data, we suggest that chicken likely simultaneously dispersed along a: 1) southern sea route; and 2) southern Himalayan and trans-Iranian/southern Central Asian route on its westward journey[17,43]. Interestingly, these two routes of dispersal have just been presented as the same two routes that rice (*Oryza sativa*) spread along at roughly the same period, ca. 2,000 years ago[98].

## Symbolic and economic prominence of the chicken in Central Asia

Historians claim that specialization in chicken farming and the focus on breeds and large-scale egg laying, as described by Columella, did not continue in Europe after the Roman period and was not resurrected until the early nineteenth century[10]. In medieval Europe the chicken took on the role of a barn-yard scavenger and may have been of some economic importance to people of lower socioeconomic status; during the early nineteenth century, the prominence of chickens in Europe again rose and specialized poultry farms developed. The situation in Central and southwest Asia appears to be different—we report the presence of chicken eggs in high ubiquity, density, and abundance at all sites that we studied for this paper through the medieval period. The rooster is prominent in Zoroastrian imagery[99]; although, so are mythical birds, such as the Sogdian hybrid bird priests and phoenix with ribbons. Zoroastrian texts make reference to the cock as associated with the god Sraosha and with the priesthood[100], the Herald of the Dawn[101] or the Guardian of Good against Evil[43]. Also, while not necessarily a chicken egg, the world-egg myth is a prominent feature in Zoroastrian belief, as with other Proto-Indo-European traditions[102].

One of the earliest and most vivid examples of this imagery in Central Asia was recovered in 2014 by the Karakalpak-Australian Expedition in the form of a wall painting in the royal contexts of Akchakhan-kala, depicting a characteristic motif of two opposing human-headed roosters in the middle of performing a Zoroastrian ritual (first century BC to first century AD[100]). The image has led some scholars to suggest stronger Central Asian roots in the origins of the rooster-worshiping cult[5,103]. The hybrid bird-priest motif represents an assistant of Sraosha and is said to be a rooster that can predict the coming of dawn, waking up the religious practitioners to fulfil their duties (Vidēvdād XVIII: 14–15, 22–23[7]). Rooster priests are prominent on Sogdian funerary couches, such as the depiction of two Zoroastrian rooster priests at a fire temple from the Shelby White and Leon Levy Collection at the Metropolitan Museum of Art (sixth century AD)[6]. There are also two rooster priests depicted in relief on the sarcophagus a Sogdian named Lord Shi, each with clear leg spurs (AD 579; Fig. 2h). On a similar sarcophagus of a Sogdian named Yu Hong (AD 592), also from Xinjiang, the tails are clearly "cock-like"[104]. Two rooster priests also flank a fire on An Jia's tympanum above the doorway to the funerary chamber[104]. Ancient Chinese depictions of funerary birds, identified as phoenixes or vermilion birds, invariably have clear features that link them in style to pheasants, whereas Central Asian funerary birds have up-turned tails and leg spurs. The clear representations of leg spurs on most of the rooster priest chimeras, verifies that they are Galliformes. In Aramaic document C-1 (line 13), from ancient Bactria, dated to the month of Kislev of the first year of Artaxerxes V (the usurper Bessus, November to December, 330 BC), among the supplies provided to Bessus as he passed from Bactria (modern Balkh) eastward to Varnu, were 30 chickens (in addition to five geese, 33 lambs, 133 sheep, one donkey, four bovines, one calf, one horse, as well as oil, wine, flour, etc[105].).

The earliest report for chickens in Central Asia currently comes from bones recovered at the site of Kyzyl Tepa, dating between the sixth and fourth centuries BC in the Syrkhandarya region of Uzbekistan[106]. Identifications of the Kyzyl Tepa zooarchaeological

remains were conducted separately by two zooarchaeologists, both of whom identified chicken bones in the assemblage. However, they represent the lowest abundance of any domesticated food animal (NISP = 12 out of 2900) and are not even presented in the summary table for domesticated animals in the study. Notably, wild goose bones are more abundant in the assemblage than chickens[106]. Other evidence for early chickens comes from a coin minted around 300 BC depicting the Satrap of Bactria, Sophytes, which is an Indian name—possibly suggesting links to the south (Fig. 2c). The coin was minted in a Greek style, depicting a rooster on one side. At least two leather applique of cockerels; have been recovered from the frozen Pazyryk tombs, which, if they are truly roosters, date to the fifth century BC and were excavated in 1929 by Gryaznov in the Bolshoy Ulagan River valley of the Altai[107]. It should be noted that, if these are proven to be true roosters, they would represent the furthest northern spread of the species this early in time; although, given the diversity of chimeric creatures represented in these frozen burials, caution should be taken. Faunal studies from the 2012 excavations at the site of Ulug-Depe in Turkmenistan resulted in the identification of chicken remains from the last phase of occupation at the site (Pre-Achaemenid and Achaemenid [Yaz II] 1100–329 BC[108]). While these remains have not yet been directly dated, it is informative to point out that they are absent from all earlier layers at the site. Chicken bones have also been recovered from the fort of Kurgansol in Uzbekistan, which is thought by the excavators to have first been established by Alexander's troops[109]. Wooden slab documents recovered from a military guard tower or postal station near the Silk Road town of Dunhuang, dating to 62 BC mention the import and export of chickens[110]. Also discovered in Xinjiang, faunal remains from Yuansha Gucheng have been reported to contain chicken bones dating to the third or fourth centuries AD[111].

Chicken bones have also been reported at the medieval Islamic capital of Shahr-e Gholgholah, located at 2600masl in the Bâmiyân Valley of Afghanistan[112]. Two roosters were also depicted flanking a fire alter in Temple B in Surkh-Kotal, Afghanistan, dating to the second century AD[99]. Two half-bird creatures were also said to have once flanked an image of Mithras on the two Buddhas of Bamiyan[113]. Further south, at the Iranian site of Dasht Qal'eh, chicken bones have been reported from layers dating to the fifth or sixth centuries AD, but other large (non-chicken) bird bones were also recovered and identified as pheasants, waterfowl, and raptors[114]. Lerner[115] compiled Sasanian (seventh and eighth centuries AD) stamp seals with rooster depictions, which she argues had apotropaic functions. One of the key cookbooks that has survived from the Golden Age of Islam, which was originally compiled in Syria for the Ayyubid rulers, is the Kitab al-Wuslah ila l-Habib fi Wasf al-Tayyibat wal-Tib (Scents and Flavors the Banqueter Favors). This book contains 635 recipes and medicinal concepts, such as presenting ways to balance the humors, an idea popular from Europe to East Asia by this time[116]. Eggs merit a special section in this book, which contains thirty-eight types of egg-based dishes. Chickens and eggs remain prominent in Central Asian imagery through the medieval period, one example is provided in Fig. 2, with a fired clay egg on display at the Arc Museum in Bukhara, dated to sometime between the tenth and twelfth centuries AD. The Hermitage also has five clay roosters in their collections from Central Asia, all from archaeological sites spanning the early Medieval period.

Linguistic evidence also suggests recent links between the bird, as it spread across Eurasia (in particular in Indo-European languages). The common Iranian word for "chicken" is *kŗka- (Avestan kahrka-, Middle Persian kark, Ossetic kark, Wakhi kₔrk, etc.), which is explained as having a phonosemantic formation meaning "cackling one", the similar Indo-European formation sometimes means chicken too (Tocharian kraṅko, Greek kérkos, old Slavic kurъ for rooster, Proto-Indo-European krenk- "to make a loud noise"), and sometimes other birds (Old Indian kŗkāra for partridge[117]). The word for rooster, such as Persian xurōs, is derived from another root meaning to cry. In many modern languages the hen retains the meaning of "bird" in general, e.g., Persian/Tajik mury, parranda (presumably from the verb paridan "to fly"), also borrowed into Uzbek and other Turkic languages, in which tovuq (Old Uyghur ťqɣyw/taqïɣu and Qarakhanid تَقاغُو /taqāɣū/) typically denoted domesticated fowl, especially hens. The word for "chicken" (Sogdian cwz'kk, Persian jujeh, etc.) is again onomatopoetic.

Chickens and chicken eggs have been an important aspect of Central Asian culture and economy for more than two millennia. Beyond a source of food, archaeological evidence has firmly established that the chicken, specifically the rooster, has long been a symbol of virility in Inner Asia, and, to the Zoroastrian faith, the rooster represents a spirit that calls at dawn to praise the triumph of light over dark in the eternal struggle between day and night. This domesticated bird's importance at the heart of the Silk Road may have facilitated its rapid dispersal across two and a half continents by roughly two millennia ago. The eggshell data we present from these sites suggest regular consumption of chicken eggs at many of the largest urban centers of medieval Central Asia, including Afrasiab, Paykend, Panjikent, and Bukhara. We also discuss finds from the medieval mountain village of Tashbulak and the late Achaemenid to Hellenistic fortified sites of Paykend and Bash Tepa. The earliest evidence provided here for non-seasonal egg laying dates to the Hellenistic period, at which time the more productive birds may have rapidly crossed Central Asia and the eastern Mediterranean. By the Roman period, specialized poultry farms existed in southern Europe and northern Africa. However, the role of the chicken in Medieval Europe switched to that of a farm-yard scavenger; whereas across Central and southwest Asia, during the Golden Age of Islam, chicken egg production appears to have remained important and continued on a large scale in cities and villages. Chickens express an impressive range developmental plasticity (a wide reaction norm) and are, therefore, easy to spread into different climates and environmental zones. The eggshell fragments from these sites at the center of the ancient trans-Eurasian trade routes illustrate: (1) that chicken eggs were likely a regular part of the diet starting in the last centuries BC; and (2) the chicken and the egg remained important in urban centers across Central Asia until at least the Qarakhanid period. As a last point, we speculate in this paper that the high abundance and ubiquity of eggshells may suggest that these birds had shifted away from a wild reproductive cycle, laying eggs over a more significant period of time.

Given the difficulties in morphologically identifying chicken bones, the highly fragmentary state they are usually preserved in, and their paucity in early contexts, peptide mass fingerprinting is well suited to clarify the domestication and dispersal narrative for the most enigmatic of the Eurasian domesticated animals. Future work, bringing together molecular methods, aDNA, oxygen isotopes for seasonality studies, and modern protocols in zooarchaeological morphology studies has potential to further clarify the questions at hand. The origins and dispersal of the chicken remains one of the great questions of animal domestication for Eurasia, and a more detailed dating and identification campaign is needed. The chicken has been an important part of the economy for more than two millennia and played a significant role in ritual and sports. As Fragner[118] (574) stated: "I suggest that a reconstruction of the path taken by the domestic fowl… throughout the ancient world would tell us much about possibilities of early human contact in areas between ancient Egypt and Babylonia, China and India; perhaps more than a single isolated coin or a piece of clay". We have added an important piece to the broader picture, but there remains much to clarify regarding this important bird.

## Methods
### Sampling and morphological identification
For most of the sites discussed in this text, systematic collection of sediment samples for archaeobotanical floatation was conducted. All sampling was conducted in cooperation with the excavation directors

of the project, and in every case we have included the excavation directors as authors on this paper. All excavation and exporting permits were obtained as part of the broader project in direct collaboration with national authorities and local researchers. This study represents a large-scale collaborative endeavor and all researchers worked together to ensure that national regulations were met. The heavy fraction of the samples was wet screened though a 1.4 mm geological sieve. The eggshells were collected from the wet screened sediments and recorded as to how many fragments per liter of sediment were recovered. For the site of Kafir Kala, eggshell fragments were handpicked by excavators during excavation, and further sediment samples have been collected but remain unfloated at present. The samples were fit into contexts using both radiocarbon dating and stratigraphic sequencing. All of the case studies in this paper are part of larger excavations and ongoing projects. The identification of eggshells was further conducted using a scanning electron microscope, which allowed us to verify that the breathing pores on the eggs matched in diameter those of modern chicken eggs in morphology and size. Specifically, the breathing pores all fell in the range between 10 and 20 μm and the thickness of the eggshells generally remained close to 300 μm. While the morphological criteria do not rule out all possible wild species, they do narrow the range of possibilities down considerably, and the features all conform to the possibility of chickens. We did not study the curvature of the fragments and attempts at determining the level of decalcification on the inner surface of the shell, an indication of the stage of development of the embryo, were inconclusive. A selection of the shell fragments was then analysed at the ZooMS laboratory at the Max Planck Institute of Geoanthropology for further identification certainty.

As an additional point, the two samples that were determined to be Anseriformes using the ZooMS method had been labled chickens based on morphology alone; although, they did not have the detailed analysis of an SEM study.

It should also be pointed out that large wild birds are prominent in many of these early zooarchaeological assemblages, complicating identifications. Other similar birds that could be mistaken for chickens in South Asia, notably Indian sites, include numerous species in the partridge (Tetraonini and Coturnicini) and pheasants and jungle fowl (Phasianini) subfamilies of the Phasianidae, the cotton teal (*Nettapus coromandelianus*) and an impressive array of waterfowl, as well as egrets (Ardeidae), ibis (Threskiornithidae), stocks (Ciconiidae), and the flamingo (*Phoenicopterus roseus*). Other large South Asian birds include the crow (*Corvus splendens*, and other relatives e.g. *C. macrorhynchos*), bulbuls (Pycnonotidae), and a wide variety of raptors, not to mention other raised exotic birds, such as the peacock, geese and ducks. In arid Central Asia and the Iranian Plateau, it would be difficult to differentiate between a chicken bone and a bone of several species of Pteroclidae (sandgrouses), chukar partridge (*Alectoris chukar*), or the houbara bustard (*Chlamydotis undulata*), all of which have a long history of bein hunter for food in this part of the world. All of these taxa are effectively excluded using peptide mass fingerprinting.

## Radiocarbon dating

All of the sites reported in this study have had separate dating campaigns; in each case, this has consisted of a significant radiocarbon dating sequence combined with ceramic seriation and in some cases verified with coins. The contexts and stratigraphy have been well worked out by a number of different excavators and are highly reliable. There have been more than 50 radiocarbon dates pulled together for this study, and a separate paper explaining and synthesizing them is in the works. In order to verify the oldest ages for these chicken eggshells, we sent two new dates, one of each of the oldest layers at Paykend and Bash Tepa to NOSAMS at Woodshole Oceanographic Institute for dating. The dates were then calibrated in the most recent calibration curve for Oxcal and presented in the results section of this paper.

## Peptide mass fingerprinting

Eggshell fragments were analyzed alongside a blank, based upon previously published methods[26,119]. Eggshell fragments of approximately 20 mg were cleaned with 400 μl 0.5 M EDTA, washed three times with 400 μl ultra-pure water and left to dry overnight. Fragments were then crushed into a powder and incubated in 200 μl NaOCL (12% w/v) for 5 days. The supernatant was discarded and the samples were rinsed five times with 200 μl ultra-pure water, then they were resuspended in methanol (100%) and left to dry overnight. Samples were demineralized in 500 μl 0.6 M hydrochloric acid (HCl) for 10 min after which 500 μl of the supernatant was transferred to a 10 kDa ultrafilter (Sartorius, Vivaspin®) and centrifuged until completely passed through the filter. 500 μl of 50 mM ammonium bicarbonate (AmBic) was then added to the ultrafilter and the samples were centrifuged a second time. The fraction that did not pass through the filter was resuspended in 200 μl AmBic. Half was transferred to a second tube, which was stored as a back-up. 11 μl of 100 mM CAA (2-chloracetamide)/100 mM TCEP (tris[2-carboxyethyl]phospine) solution was added to the remaining sample, and then they were digested with 1 μl 0.4 μg/μl of trypsin solution (Pierce™ Trypsin Protease, Thermo Scientific) for 18 h at 37 °C. Following enzymatic digestion, peptides were purified and concentrated using 100 μl C18 resin ZipTips (Pierce™ C18 Tips, Thermo Scientific) with conditioning and eluting solutions composed of 50% acetonitrile (v/v) and 0.1% TFA and a lower hydrophobicity wash buffer of 0.1% TFA. Peptides were eluted in 50 μl conditioning solution.

The samples were spotted in triplicate onto an MTP AnchorChip 384-target plate, together with matrix solution (10 mg α-cyano-4-hydroxycinnamic in 7 mL 85% acetonitrile [ACN]/0.1% trifluoroacetic acid [TFA]) and analysed using an Autoflex Speed LRF matrix-assisted laser desorption/ionization-tandem time of flight mass spectrometer (MALDI-TOF-MS, Bruker Daltonics) with a smartbeam-II laser. A SNAP averaging algorithm was used to obtain monoisotopic masses (C: 4.9384, N: 1.3577, O: 1.4773, S: 0.0417, H: 7.7583). Resulting spectra were examined manually using the open-source software mMass[120] using peak picking with a signal to noise ratio of 3.0. Observed peaks were matched to a list of published taxon-specific $m/z$ values[26,121–123].

In order to confirm taxonomic identification, one eggshell sample with a high-quality MALDI spectra, identified as chicken (sample CP565, Afrasiab), was selected for further analysis using liquid chromatography with tandem mass spectrometry (LC-MS/MS) to obtain peptide sequence data. 20 μl of the sample extract was dried down and sent for LC-MS/MS analysis at the Functional Genomics Center Zurich. LC-MS/MS was conducted using a Q-Exactive HF mass spectrometer (Thermo Scientific) coupled with an ACQUITY UPLC M-Class system (Waters AG). Solvent composition at the two channels was 0.1% formic acid for channel A and 0.1% formic acid, 99.9% ACN for channel B. Column temperature was 50 °C. For each sample, 4 μl of peptides were loaded on a commercial MZ Symmetry C18 Trap Column (100 Å, 5 μm, 180 μm × 20 mm, Waters) followed by nanoEase MZ C18 HSS T3 Column (100 Å, 1.8 μm, 75 μm × 250 mm, Waters). The peptides were eluted at a flow rate of 300 nL/min by a gradient from 5 to 40% B in 120 min and 98% B in 5 min. The column was cleaned after each run with 98% solvent B for 5 min and holding 98% B for 8 min prior to re-establishing loading condition. The mass spectrometers were operated in data-dependent mode performing HCD (higher-energy collision dissociation) fragmentation on the 12 most intense signals per cycle. Full-scan MS spectra (300–1500 $m/z$) were acquired at a resolution of 120,000 at 200 $m/z$ after accumulation to a target value (AGC) of 3,000,000, while HCD spectra were acquired at a resolution of 30,000 using a normalized collision energy of 28 (maximum injection time: 50 ms; AGC 10,000 ions). Unassigned singly-charged ions and ions were excluded. Precursor masses previously selected for MS/MS measurement were excluded from further selection for 30 s, and the exclusion window was set at 10 ppm. The samples were

acquired using internal lock mass calibration on *m/z* 371.1012 and 445.1200.

Byonic v.3.2.0 (Protein Metrics Inc[124].) and Mascot[125] were used to analyse the LC-MS/MS data. Using Byonic, product ion spectra were searched against a reference database consisting of all sequence data (duplicates removed) in Swissprot (download May 13, 2022); the entire proteomes of *Anas plathyrhynchos* ssp. *plathyrhynchos* (UP000016666), *Aquila chrysaetos* ssp. *chrysaetos* (UP000472275), *Cotornix coturnix* (UP000694412), *Bambusicola thoracicus (Perdix thoracica)* (UP000237246), *Corvus brachyrhynchos* (UP000052976), *Gallus gallus* (UP000000539), *Haliaeetus albicilla* (UP000054379), *Numida meleagris* (UP000243875), *Opisthocomus hoazin* (UP000053605), *Phasianus colchicus* (UP000472261), and *Pterocles gutturalis* (UP000053149); all proteins available in UniProt for the galliforms *Alectoris chukar*, *Lagopus lagopus*, *Lagopus muta*, *Lyrurus tetrix*, *Perdix dauurica*, *Tetrao urogallus*, *Tetraogallus himalayensis*, and *Tetrastes bonasia*; and all sequences for Aves species from UniProt and NCBI for the eggshell proteins clusterin, osteopontin, ovalbumin, ovocleidin-17, ovocleidin-116, and ovotransferrin using the following parameter settings: fragment mass error: 20 ppm; precursor mass error: 5 ppm; semi-specific tryptic digestion; 2 missed cleavages allowed; mass changes: 2 common, 1 rare; fixed: carbamidomethyl on cysteine (C); common: oxidation on histidine (H), methionine (M), proline (P) and tryptophan (W), dioxidation on M and W, trioxidation on C, deamidation on asparagine (N) and glutamine (Q), pyro-Glu on N-term Q; rare: ammonia-loss on N-term C, pyro-Gly on N-term glutamic acid (E); no sequence variations allowed, wildcard search disabled.

The proteins present in the sample were identified by protein FDR 1%, log probability ≥ 5, at least 2 peptides with a PEP2D score lower than 0.01. The masses of all observed MALDI *m/z* peaks were compared to the list of peptides identified in these proteins, peptide matches were reported if the PEP2D score was lower than 0.01. These sequences were compared to published markers, checked for uniqueness using NCBI Blast and mapped back onto the corresponding protein sequences using Geneious Prime 2020.1. MALDI peaks were only considered for identification if only one peptide was present in the LC-MS/MS data at that mass and based upon their uniqueness to taxonomic level. A final list of all peptides and proteins identified in the sample can be found in Supplementary Table 1 and Extended Data Table 3.

Spectral data was also searched with Mascot against the Swissprot (downloaded 17 August 2023) combined with the custom created database of egg and eggshell proteins from Eurasian bird species (Supplementary Table 2). Search settings included carbamidomethylation of cysteine (C) as a fixed modification, and deamidation of asparagine (N) and glutamine (Q), and oxidation of methionine (M) as variable modifications. Trypsin was selected as the enzyme, and the peptide mass tolerance was set at 10ppm, with the fragment tolerance at 0.01, with a allowances for a single carbon isotopic shift. The instrument was selected as ESI-QUAD-TOF. These resulting data were further filtered using MS-MARGE, which excluded any peptides with an e-value above 0.01, and all proteins supported by fewer than two peptide spectral matches. Peptide and protein FDR were both <0.01.

### Reporting summary
Further information on research design is available in the Nature Portfolio Reporting Summary linked to this article.

## Data availability
The MALDI-ToF-MS spectra generated in this study have been deposited on Zenodo (doi:10.5281/zenodo.4084517)[126]. The MS/MS data files are available on ProteomExchange under accession code PXD031493 and were uploaded through MassIVE (MSV000088794, doi:10.25345/C5HK35)[127]. The source data for Fig. 4 can be found in these datasets. All specimens not used for molecular analysis have been stored at the Max Planck Institute of Geoanthropology, and are properly curated. All other data is available in the Supplementary Information. Correspondence and requests for materials should be addressed to Robert N. Spengler III (spengler@shh.mpg.de).

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

## Acknowledgements

B. Demarchi provided proteomic reference data for chicken eggshell. Research funds and support were provided by the Max Planck Society and the European Research Council, grant number 851102, Fruits of Eurasia: Domestication and Dispersal (FEDD).

## Author contributions

R.N.S. conceived and designed the study. R.N.S., H.R., B.B.M., S.Sc., K.A., M.v.B. and R.D.M. processed sediment samples and collected eggshell fragments. C.P. and K.K.R. conducted laboratory analysis. C.P., K.K.R. and S.Wi. analyzed and interpreted palaeoproteomic data. S. St., F.M., M.Si., H.R., A.B., M.F., S.K., M.Sh., T.H., F.K., AO., S.Wa. and P.L. provided samples and ran project excavations. R.F., N.B., T.H., B.H., and R.N.S. interpreted the overall data and assisted in the construction of the discussion. C.P. and R.N.S. prepared figures. C.P. curated data. C.P. and R.N.S. wrote the paper, with critical input from all authors.

## Funding

## Competing interests

The authors declare no competing interests.

## Additional information

[1]Department of Archaeology, Max Planck Institute of Geoanthropology, 07745 Jena, Germany. [2]Department of Anthropology, Harvard University, Cambridge, MA 02138, USA. [3]Institute of Evolutionary Medicine, Medical Faculty, University of Zurich, 8057 Zurich, Switzerland. [4]Institute for the Study of the Ancient World, New York University, New York City, NY 10028, USA. [5]Domestication and Anthropogenic Evolution Research Group, Max Planck Institute of Geoanthropology, 07745 Jena, Germany. [6]Faculty of Arts, Masaryk University, Nováka 1, 602 00 Brno-střed, Czech Republic. [7]Department of Bioarchaeology, Faculty of Archaeology, University of Warsaw, ul. Krakowskie Przedmieście 26/28, Warszawa 00-927, Poland. [8]Climate Change and History Research Initiative, Princeton University, Princeton, USA. [9]National Center of Archaeology, Uzbekistan Academy of Sciences, Tashkent 100000, Uzbekistan. [10]Samarkand Institute of Archaeology, Agency for Cultural Heritage, 1000060 Samarkand, Uzbekistan. [11]Berlin-Brandenburg Academy of Sciences and Humanities, 10117 Berlin, Germany. [12]Department of Linguistics, University of Vienna, Wien 1090, Austria. [13]Department of Archaeology, Samarkand State University, Samarkand City 140104, Uzbekistan. [14]Department of Anthropology, Washington University in St Louis, St Louis, MO 63130, USA. [15]School of Cultural Heritage, Northwest University, Xi'an 710069, China. [16]Institute of History, Archaeology and Ethnography named after Ahmad Donish of the Academy of Sciences of Tajikistan, Dushanbe, Tajikistan. [17]Department of Islamic and Middle Eastern Studies, The Hebrew University of Jerusalem, Mt Scopus, 91905 Jerusalem, Israel. [18]New Uzbekistan University, 54 Mustaqillik Ave, Tashkent 100007, Uzbekistan. [19]Department of Archaeogenetics, Max Planck Institute for Evolutionary Anthropology, 04103 Leipzig, Germany. [20]Department of Anthropology, University of Arkansas, AR 72701 Fayetteville, USA. [21]New York University Abu Dhabi, Abu Dhabi, United Arab Emirates. [22]State Hermitage Museum, St Petersburg 190000, Russia. [23]Centre de Recherche et d'Enseignement des Géosciences de l'Environnement, Aix-Marseille Université, CNRS, IRD, INRAE, 13545 Aix-en-Provence, France. [24]Institute of Archaeological Science, University of Tübingen, 72070 Tübingen, Germany. [25]Griffith Sciences, Griffith University, Nathan, QLD 4111, Australia. [26]School of Social Science, The University of Queensland, Brisbane, QLD 4071, Australia. [27]School of Archaeology and Museology, Peking University, Beijing 100871, China. ✉e-mail: spengler@shh.mpg.de

