## [Peer Review File · Nature Communications]

Archaeological and molecular evidence for ancient chickens in Central AsiaReviewers' Comments:

Reviewer #1:

Remarks to the Author:

The early dispersal and social role for chicken are still under multidisciplinary discussion. In this study, Peters, et al. adopted peptide mass fingerprinting to identify chicken eggs from 12 Central Asian sites spanning 200 BC to AD 1220. In addition to presenting a practical approach in chicken archaeology, the authors provide solid evidence for early chicken as well as egg consumption in southern Central Asia. The study has potential to attract wide concerns from archaeology, anthropology, and animal genetics.

The conclusions and claims are well-supported by the archaeological evidence. Nevertheless, the manuscript still can be improved. Especially, how to integrate the current study into the context of chicken domestication and dispersal should be well-addressed. I have some comments as following.

Major Comments:

The color of egg is informative. In chicken genetics, series of studies have been performed to map the genes. I suggest the authors to trying to recover the information for egg colors (e.g., white, brown, or blue). It will provide insights into ancient chickens.

Line276-294: The authors tried to compare their results with those from previous ancient DNA studies which suggested that the derived TSHR gene occurring in Europe was late to Middle Ages. The authors may consider and explain the difference, especially given that the small sample size in ancient DNA studies.

Line375-385: The authors presented the linguistic evidence to indicate the recent link of chicken across Central Asia. It should be noted that, the languages (even for Avestan) analyzed are recently derived type. Is it possible to analyze more Indo-European languages, such as Tocharian, Anatolian, Armenian, and Slavic? Given the divergence time of The results will verify the conclusion of recent

The authors cited the paper by West and Zhou (Ref15). In that paper, West and Zhou proposed the norther route from northern China across Russia to West Europe for chicken dispersal after domestication. The authors may consider to have some discussion about the origin of Central Asian chickens. Were the chickens likely from South Asia or East Asia via the ancient Silk Road (Steppe Road)?

Minor Comments:

Line63: The representative papers for genetic and molecular methods (e.g., Cell Res. 2020;30:693-701) should be cited here.

Line97: Mithras in Germany. In Germany of Persia/Iran should be checked.

Line122: In "this study fills and important", delete and.

Line138: The reference 13 is not reported the archaeological sites in Japan and Korea.

Line164: It is better to provide more information about Lord Shi.

Reviewer #2:

Remarks to the Author:

The article is associated with the history of chicken egg exploitation in southern Central Asia. To reveal the exploitation of chicken egg in the region, authors studied eggshell fragments from 13

archaeological contexts in the southern Central Asia, using morphological and peptide mass fingerprinting approaches. Based on results, authors discussed that chickens have been part of the subsistence economy in southern Central Asia for roughly 2200 years and that the loss of seasonal egg laying was the main driver for the rapid dispersal across Eurasia and northeast Africa. Because the results contain a lot of new findings about the history of chicken egg exploitation, the contents are worth to be published. There are, however, at least three major problems, I cannot recommend to the editor to accept the paper as is.

Major comments:

1. Lack of reliable identification for eggshell:

Authors conducted morphological and peptide mass fingerprinting identification in the manuscript. However, results of morphological identification are not fully explained. They described the size and morphology of the archaeological eggshell breathing pores as consistent with modern chicken eggs (L581-584), but that description is not sufficient for those eggshells to identify chicken eggshells. The number of morphologically identified samples is unknown (this is also a problem), while only 16 eggshells from 4 sites were identified as chicken by peptide mass fingerprinting. Other archaeological eggshell fragments may also belong to other species, as two of the eggshells were identified as 'Anseriformes' by peptide mass fingerprinting. Therefore, it is inappropriate to argue that the results presented the paper indicate that chicken eggshells were found at 12 sites. On the other hand, the samples subjected to peptide mass fingerprinting also do not appear to provide sufficient evidence to distinguish them from chicken (*Gallus gallus*) eggshells. Although LC-MS/MS is applied to one eggshell sample, it is logically impossible to identify chickens rather than other local Phasianidae birds without data on local Phasianidae species eggshell proteins. Ring-necked pheasants are the most closely related to chickens in the local Phasianidae species, but this does not guarantee that chickens and ring-necked pheasants are the most closely related for all peptide fragments. Furthermore, even if pheasants and chickens can be distinguished according to PMF markers found in previous studies, can chickens be distinguished from other *Gallus* junglefowl? To identify it as *Gallus gallus*, it must be shown that it is not another *Gallus* junglefowl. Revision is required.

2. Lack of reliable dating for eggshell:

As reviewed by the authors, reliable identification and direct dating of chicken bones is essential to uncovering the presence of chickens at the site at that time. Is there evidence that the eggshells analyzed in this study were not contaminated from a later period? In addition, the inconsistency in sample ages undermines the reliability of the data. For example, the date of samples from Bash Tepa is ca 3rd century BC- 1st century AD in the text (L112-113) but 200-1 BC in Table 1 and 400-0 BC in Extended Data Table 1. Revision is required.

3. Lack of the evidence for non-seasonal egg laying:

The idea is very interesting. However, the evidence is, in my sense, very weak. The only evidence shown by authors is "artefacts recovered in many different archaeological contexts are more likely to represent frequent-occurrence events, as opposed to seasonal or semi-annual events" (L240-242). Judging from the references given there, only two closely related authors mention this pattern in their books. Is this a well-accepted proof for archaeobotanists? Also, how much ubiquity does it take to say that it is not a seasonal or semi-annual event? In Paykend 1, the ubiquity is 30%. Is this enough to say frequent-occurrence events? I think there is a big gap between frequent-occurrence events and non-seasonal egg laying, and common food over a greater period of the season (L247-248).

Minor comments:

L77-79 "Further blurring... or heavily hunted":

Why is the use of other birds relevant to understanding the early dispersal of chickens? Please explain.

L82-84 "In this manuscript ,... AD1220":

What do you mean by "success" here? Are the results of morphological analysis consistent with the results of peptide mass fingerprinting? Need an explanation in "Results" section.

L84-85 "The lack of ...archaeological sites":

Are there archaeological sites from the same era or earlier that attempted to collect eggshells with a sieve like this research? Please explain with adequate references. The "a rapid rise of egg-laying and chicken rearing" claim is irrelevant if there has been no such sieve sampling in earlier times.

L137-138 "the bird is... AD 300 13)"

The Yayoi period is inadequate for Korea. Ref 13 would be inadequate here.

Figure 1: Please show the site names in the caption. For archaeological sites in Uzbekistan and Tajikistan, indicate them numerically on the map and indicate the site name in the caption.

L174-176 "Additionally, ... evidence for absence)":

The description is not a result of this study.

L179:

Are you studied samples from 13 sites?

L185-187:

In Paykend 1, the ubiquity is 30%. It is obviously less than 50%.

Table 1:

Please explain what is the "Number of Samples".

L217-223:

Were the samples identified as Anseriformes considered to be other than chickens in morphological examination?

L405-406 "Chicken express... environmental zones":

References are required.

Reviewer #3:

Remarks to the Author:

- What are the noteworthy results?

The article addresses the cultivation of chicken in Central Asia. A major strength is the molecular techniques and eggshell analysis that is presented to reassess chicken domestication. The review provides extensive detail of sites, dates, evidence found, and commentary regarding the acceptability of the evidence and reasons why. This is a huge manuscript when the supplemental material is considered and will be useful to researchers for many years.

- Will the work be of significance to the field and related fields? How does it compare to the established literature? If the work is not original, please provide relevant references.

This article is the most extensive review I have ever seen regarding the topic across existing literature. The information presented is significant and contributes greatly to the topic in archaeology, poultry science, molecular biology, and veterinary sciences. The work is entirely original, although the preprint of the manuscript is available also as Spengler et al.

Spengler III, R. N., Peters, C., Richter, K. K., Mir Makhamad, B., Stark, S., Fernandes, R., et al. (2022). When did the chicken cross the road: archaeological and molecular evidence for ancient chickens in Central Asia. Research Square, 1340382/v1. doi:10.21203/rs.3.rs-1340382/v1.

- Does the work support the conclusions and claims, or is additional evidence needed?

Yes. No additional evidence is warranted as extensive details are presented

- Are there any flaws in the data analysis, interpretation and conclusions?- Do these prohibit publication or require revision?

None that I can identify

- Is the methodology sound? Does the work meet the expected standards in your field?

Yes, the methodology regarding their approach to reviewing the sites and conducting the eggshell analysis is sound and explained in great detail exceeding expected standards.

- Is there enough detail provided in the methods for the work to be reproduced?

Absolutely. The information regarding research steps and the material and equipment used will allow the work to be reproduced.

General Comments

- The research presented results from a huge endeavor by the authors. The manuscript could basically be divided into two articles even. The first about the eggshell analysis and second about their review of the literature and evidence
- In my view, the points that the authors make are often hidden in their extensive review in the Discussion. This is often unavoidable when reviewing so much information. Perhaps a review of these points could be made in the Conclusion

Introduction 62-151

- 78 The author may consider rewriting the sentence to use the term "husbandry" instead of "cultivated"
- 82 suggest keeping all information about the manuscript together. Move the sentence "In this manuscript..." to the beginning of 105
- 89-104 suggest moving this paragraph and starting a new one after the sentence in 79-82

Dissemination across the Ancient World 128-151

- 142 suggest more direct and concise sentences such as "Debate of early chicken spread focus on the reliability..."
- 149-150 not really sure if (Figure 1) is necessary as this could be referred to in an added reference
- 151 suggest adding an exemplary reference ("textual source") at the end of the sentence

Results

Archaeological Eggshells 170-191

- 172 add "(Figure 1) after "in this study,"
- 175 for conciseness, phrases such as "bearing in mind" should be avoided. Use "considering that the..." There may also be other sentences in the manuscript that could be edited to remove/change similar phrases
- 179-180. I believe there are 13 sites in Table 1

Discussion 230-285

Evidence for Non-Seasonal Egg Laying 232-294

- 234 says "We present evidence for ancient chicken eggshells from 13 different archaeological settings, spanning a period of a millennium (Table 1)." In Table 1, 13 sites are listed.
- 250-253 suggest rewriting as "The resulting evidence suggests that these chickens expressed shifts in reproduction from the wild and were producing eggs at a regular rate for a significant part of the year; although, these data do not allow us to specify the duration or abundance of laying."

Symbolic and Economic Prominence of the Chicken in Central Asia 296-223

- 298-310 This section could be improved by beginning with the reporting and interpretations by the authors rather than other studies to make it stronger. The specific results of the study and the support the evidence provides become hidden in the extensive review.
- 335-337 Not sure what the need is for including the names of the zooarchaeologists who identified remains

Conclusions 387-428

- 406 perhaps say " The eggshell data we present from these sites..."

References 431 566

- I leave the checking and accuracy to the copy editors reviewing the manuscript.

Methods 569-672

- Nice details are presented in the Methods to explain the procedures for peptide mass fingerprinting
- 584-586 Should the sentence "The shells were mostly white in color; although, some from Tashbulak appeared to have speckling, which may represent post depositional discoloration or possibly a distinct variety of chicken." be placed in the Results section?

Methods References 675-759

- I leave the checking and accuracy to the copy editors reviewing the manuscript.

Figures and Tables

Figure 1. Should clarify the placement of points 45-49, and it is confusing between 48 and 49 and the cluster of 45-47 contains four points. This is always a problem in a distribution figure such as this.

Figure 2 Pictures e and f could benefit from adding size bars similar to a and b

Table 1: not sure how Nature Communications handles empty table cells

Supplemental Material

Extended Data Table 1: Key Archaeological Sites Supporting the Westward Spread, Linked to Figure 1.

- Listing and information are fine

Extended Data Table 2: Sample numbers, site and context information, and final identification of the eggshell fragments included in this study.

- Listing and information are fine

Extended Data Table 3: Proteins identified in sample CP565 (Afrasiab) with a LogProb ≥ 5 , and at least 2 unique peptides.

- Listing and information are fine

Supplementary Discussion 1: The Chicken Domestication Debate

- This section is an amazing review of the status of chicken domestication. But with all the information presented, the stances of the authors are not easily found or discerned. Statements at the beginning of each section or paragraph would make the authors' interpretation and stance stand out.
- May be useful to provide the names of other researchers in the first section similar to your references of Best, Peters, West & Zhou, and others in the later paragraphs. It is nice to be able to keep track of the researchers and their debates without looking up the reference number

Widely Conflicting Claims

- Which claim is most acceptable to the authors?

Dispute over the Chinese Center of Origin

- What evidence and date are the most acceptable to the authors?

Lack of Resolution

- What dispersion route is most acceptable to the authors?

Supplementary Discussion 2: Dissemination across the Ancient World

- Can the authors state which study they consider to provide the best evidence of dissemination?

Supplementary Discussion 3: The Sites Discussed in this Manuscript

- Suggest revising the title of this section to: "Supplementary Discussion 3: Sites containing Eggshell Discussed in this Manuscript"
- Suggest providing the sites in the order they are presented in Table 1
- Ming Tapa is not found in Table 1

Supplementary Discussion 4: Data and Results of Peptide Mass Fingerprinting of the Eggshells

- Is this the take-home message for this section: "The closest relative to chicken present in Uzbekistan, ring-necked pheasant, can be distinguished from chicken using published markers, and we find only evidence corresponding to chicken specific peptides in the LC-MS/MS, so, even without having references from local wild taxa, we can securely use these MALDI peptide markers to identify a sample as chicken as opposed to local, wild taxa." Although it is a matter of presentation style in scientific writing, the authors may consider placing their definitive statements at the beginning of a section rather than the end. I only recommend this, because in my recent two article submissions this point was raised to improve the manuscript.

References

- 92 Should it be "Brown"?
- Ref 117 should be Eda not Edu

REVIEWER COMMENTS

Reviewer #1 (Remarks to the Author):

The early dispersal and social role for chicken are still under multidisciplinary discussion. In this study, Peters, et al. adopted peptide mass fingerprinting to identify chicken eggs from 12 Central Asian sites spanning 200 BC to AD 1220. In addition to presenting a practical approach in chicken archaeology, the authors provide solid evidence for early chicken as well as egg consumption in southern Central Asia. The study has potential to attract wide concerns from archaeology, anthropology, and animal genetics.

The conclusions and claims are well-supported by the archaeological evidence. Nevertheless, the manuscript still can be improved. Especially, how to integrate the current study into the context of chicken domestication and dispersal should be well-addressed. I have some comments as following.

We thank the reviewer for putting time into reading and thinking about our manuscript and for providing insightful comments.

Major Comments:

The color of egg is informative. In chicken genetics, series of studies have been performed to map the genes. I suggest the authors to trying to recover the information for egg colors (e.g., white, brown, or blue). It will provide insights into ancient chickens.

We agree, this is an important point – we moved the sentence about the egg color from the SI section to the main text and also added another image of a pile of the eggshells from the Bukhara site so the readers can judge the colors for themselves. In short, all of the eggshells are white and appear to have always been white, except for a few of the shells from Tashbulak, some of which have a faint specking pattern, which could either be genuinely part of the egg or a result of post-depositional staining. The new photos of the shell fragments that we have added are in Figure 2; these should allow the reader to make their own judgements regarding color.

Line276-294: The authors tried to compare their results with those from previous ancient DNA studies which suggested that the derived TSHR gene occurring in Europe was late to Middle Ages. The authors may consider and explain the difference, especially given that the small sample size in ancient DNA studies.

The reviewer is raising another interesting point here; although, we (the authors) have debated whether or not to leave this comment in the manuscript, as there is not much that we can say about it here. We would rather not dive into speculation on the topic, especially seeing that follow-up genetic studies will eventually be conducted by Greger Larson and his lab – we have now collected chicken bone specimens from these same sites for his team to work with. Over the course of the next couple years this project will expand and those questions will, hopefully, be addressed.

Line375-385: The authors presented the linguistic evidence to indicate the recent link of chicken across Central Asia. It should be noted that, the languages (even for Avestan) analyzed are recently derived type. Is it possible to analyze more Indo-European languages, such as Tocharian, Anatolian, Armenian, and Slavic? Given the divergence time of The results will verify the conclusion of recent

We actually have Tocharian and Slavic already noted, and as we already point out, they are not exactly helpful in tracing the spread of the chicken, as all of these words are phonosemantic formations, meaning “cackling one” or “to make a loud noise”. Basically, everyone all through the past two millennia of human history agrees that chickens make an annoying sound. In hopes to please this reviewer, we have also added: Proto-Tocharian: kränk-ān- and the theoretical Proto-Indo-European: krenk-

The authors cited the paper by West and Zhou (Ref15). In that paper, West and Zhou proposed the norther route from northern China across Russia to West Europe for chicken dispersal after domestication. The authors may consider to have some discussion about the origin of Central Asian chickens. Were the chickens likely from South Asia or East Asia via the ancient Silk Road (Steppe Road)?

As we discuss in detail, the SI section, the claims of West and Zhou for early chickens in China are wrong, which means that all secondary arguments that they make based on their erroneous results are also wrong. Without the early Chinese center of origin, there simply is no grounds to even argue for a northern route of dispersal – we present no data for chickens north of Ferghana and there are no data points for chickens on the steppe or in Russia until the early historic period (well, except for the textile preserved in a Pazyryk grave from the Altai, which we note but are cautious about). We are not sure that there is really much of a discussion to have regarding the northern route of dispersal, as there is no data for it and we present no new data that could add to such a claim.

As the corresponding author has spent his career, thus far, exploring the northern dispersal routes, he would have been happy to argue for one here, but the data simply do not support it.

Minor Comments:

Line63: The representative papers for genetic and molecular methods (e.g., Cell Res. 2020;30:693-701) should be cited here.

We discuss this paper in detail in the SI section, but we are happy to also cite it here – citation added.

Line97: Mithras in Germany. In Germany of Persia/Iran should be checked.

Well, it is a Roman site – thanks, we have checked, and verified that our wording is clear. We also added the “Iranian”, as the reviewer maybe suggesting we do here.

Line122: In “this study fills and important”, delete and.

We thank the reviewer for catching our error here, we changed the 'and' to 'an'.

Line138: The reference 13 is not reported the archaeological sites in Japan and Korea.

We are grateful for the reviewer for catching this, somehow this citation had been switched; we have now fixed it.

Line164: It is better to provide more information about Lord Shi.

Yes, we agree that using his name without an explanation of who he was is confusing. We have added some text to try to clarify – both in the figure caption and in the text.

We thank Reviewer I for her/his detailed comments and we are grateful for the time they put into the paper. We hope that they are now happy with our responses to their queries.

Reviewer #2 (Remarks to the Author):

The article is associated with the history of chicken egg exploitation in southern Central Asia. To reveal the exploitation of chicken egg in the region, authors studied eggshell fragments from 13 archaeological contexts in the southern Central Asia, using morphological and peptide mass fingerprinting approaches. Based on results, authors discussed that chickens have been part of the subsistence economy in southern Central Asia for roughly 2200 years and that the loss of seasonal egg laying was the main driver for the rapid dispersal across Eurasia and northeast Africa. Because the results contain a lot of new findings about the history of chicken egg exploitation, the contents are worth to be published. There are, however, at least three major problems, I cannot recommend to the editor to accept the paper as is.

Major comments:

1. Lack of reliable identification for eggshell:

Authors conducted morphological and peptide mass fingerprinting identification in the manuscript. However, results of morphological identification are not fully explained. They described the size and morphology of the archaeological eggshell breathing pores as consistent with modern chicken eggs (L581-584), but that description is not sufficient for those eggshells to identify chicken eggshells. The number of morphologically identified samples is unknown (this is also a problem), while only 16 eggshells from 4 sites were identified as chicken by peptide mass fingerprinting. Other archaeological eggshell fragments may also belong to other species, as two of the eggshells were identified as 'Anseriformes' by peptide mass fingerprinting. Therefore, it is inappropriate to argue that the results presented the paper indicate that chicken eggshells were found at 12 sites. On the other hand, the samples subjected to peptide mass fingerprinting also do not appear to provide sufficient evidence to distinguish them from chicken (*Gallus gallus*) eggshells. Although LC-MS/MS is applied to one eggshell sample, it is logically impossible to identify chickens rather than other local Phasianidae birds without data on local Phasianidae species eggshell proteins. Ring-necked pheasants are the most closely related to chickens in the local Phasianidae species, but this does not guarantee that chickens and ring-necked pheasants are the most closely related for all peptide fragments. Furthermore, even if pheasants and chickens can be distinguished according to PMF markers found in previous studies, can chickens be

distinguished from other *Gallus junglefowl*? To identify it as *Gallus gallus*, it must be shown that it is not another *Gallus junglefowl*. Revision is required.

We agree with the reviewer that it is important to be able to uniquely identify the eggshell remains. In an ideal world we would have reference material for every locally present species in the Galliformes family, but this was not possible. Therefore, we created a database that includes all available protein data from all seven Phasianidae species (subfamilies *Perdicinae* & *Phasianinae*) that are locally present in Uzbekistan (see Supplementary Discussion 4).

We know from previously published data that there are differences between the MALDI spectra for chicken, pheasant, quail and other species in the Galliformes Order (Presslee et al. 2017). For the LC-MS/MS searching we knew already from the MALDI data that the sample belonged to a galliform. Therefore, we were focused on including galliform species. We included some additional species that were closely related to species locally present based upon data from avibase that had high quality data available, unfortunately many of the most obvious species do not have published proteomes. In order to combat that the database also includes key eggshell proteins that are available from all species. This single analysis represents all eggshells identified as chicken, and they all have the same set of peptide markers identified from their MALDI spectra which correspond to previously published chicken markers also verified by LC-MS/MS.

Finally, we have checked all peptides for uniqueness to chicken using blast against the larger NCBI database. We are confident that the peptides we are seeing in high abundance in the MALDI derive uniquely from chicken and not from other locally present wild galliform species that lacked published MALDI markers, which was the largest concern given the published MALDI markers that had already clearly identified the sample as *Gallus gallus*. We cannot exclude that the eggshell does not derive from *Gallus junglefowl*. However, the geographic range of this species falls far out of the current region of study. Additionally, the number of eggshell fragments recovered from the sites makes it highly unlikely these eggshells were imported.

To address the reviewer's concerns, we have added these last three sentences to Supplementary Discussion 4.

Presslee, S. *et al.* The identification of archaeological eggshell using peptide markers. *Sci. Technol. Archaeol. Res.* **3**, 89-99, doi:10.1080/20548923.1424300 (2017).

We have attempted to make it clear throughout the text that we do not believe that the morphological criteria for identification is not, in and of itself, enough to confidently say that these are chicken eggshells. However, we cite several studies that have used these morphological criteria to differentiate between chickens and other birds in the past. We are seeking a more reliable approach in this manuscript and we are rather confident that our current approach, which utilizes ZooMS (a highly reliable methods), descriptive morphology, art historical sources, historical evidence, and to a lesser extent zooarchaeology (note that proper zooarchaeological studies are underway at several of these sites and they are also finding clear evidence for chickens), achieves a reliable conclusion. We are very happy to accept that there may be non-chicken eggshells (possibly from domesticated geese) mixed into the assemblages, but it is beyond the scope of our study to analyze all of these eggshells. Follow-up aDNA studies are planned with a team under Greger Larson's direction, so over the next several years

there will be additional clarity. Given that it has taken nearly a decade to collect this data and more than four years to process the eggshells and compile the historical and art historical sources, we hope that this reviewer will find our manuscript worth publishing.

2. Lack of reliable dating for eggshell:

As reviewed by the authors, reliable identification and direct dating of chicken bones is essential to uncovering the presence of chickens at the site at that time. Is there evidence that the eggshells analyzed in this study were not contaminated from a later period? In addition, the inconsistency in sample ages undermines the reliability of the data. For example, the date of samples from Bash Tepa is ca 3rd century BC- 1st century AD in the text (L112-113) but 200-1 BC in Table 1 and 400-0 BC in Extended Data Table 1. Revision is required.

We appreciate the reviewer's concerns regarding the dating, and we have paid for additional rushed radiocarbon dates. These new dates clearly illustrate the oldest ages of our chickens – we added the dates to the paper along with clarification of the discussion. We had not run radiocarbon dates before, as our broader project has run more than two dozen dates on seeds from these sites and the chronologies are fixed with many more radiocarbon dates run by the project excavators, along with a very well-worked ceramic and numismatic seriation. Bash Tepa, in particular, has a highly detailed chronology sorted out through four-years of excavation, including an additional nearly two dozen dates, run by Soren Stark and his team. We absolutely do not dispute the possibility that some of the eggshell fragments have been turbated – we have enough experience with unexpected archaeobotanical material being stratigraphically out of place in sites – however, when dealing with thousands of eggshells from dozens of contexts at 13 different sites, the chances that everything is out of place is extraordinarily improbable.

We hope the reviewer finds the new dates suitable to put all concerns at ease, and again, we thank the reviewer for raising such concerns.

3. Lack of the evidence for non-seasonal egg laying:

The idea is very interesting. However, the evidence is, in my sense, very weak. The only evidence shown by authors is “artefacts recovered in many different archaeological contexts are more likely to represent frequent-occurrence events, as opposed to seasonal or semi-annual events” (L240-242). Judging from the references given there, only two closely related authors mention this pattern in their books. Is this a well-accepted proof for archaeobotanists? Also, how much ubiquity does it take to say that it is not a seasonal or semi-annual event? In Paykend 1, the ubiquity is 30%. Is this enough to say frequent-occurrence events? I think there is a big gap between frequent-occurrence events and non-seasonal egg laying, and common food over a greater period of the season (L247-248).

We recognize that this is a logic argument, and we have nothing more to add to it at this point. If the reviewer is unwilling to accept our argument, we simply cannot provide more proof at present. However, we strongly believe that this reasoning is well-accepted in the field of archaeobotany. Additionally, we simply cannot imagine an alternative option, we would go so far as to say that we find it highly improbable that our reasoning is wrong. Despite our confidence, we have attempted to remain as cautious as possible in the wording and have tried to avoid specifics, such as how many eggs a year a chicken might have been laying at this time or how long the laying season might have been. That said, a wild laying state would suggest that one brood a year, or roughly six chickens, was the norm; finding

thousands of shell fragments across dozens of different anthropogenic contexts seems unlikely given the wild laying state. Further backing up are discussion, are historical sources from only a few centuries later in time that clearly discuss non-seasonal egg laying patterns among chickens in the Classical world. We appreciate the reviewer's concerns here, but we hope that they will be lenient on this point, as we do think it is important to leave it in the manuscript, as some discussion of the cultural aspects of this study is important to us.

As a side note, we are in the process of brainstorming a check for this theory that involves isotopes from eggshells and banded-sampling of isotopes on ungulate teeth from the same context. While there will be many complicating factors in such a study, it may show, to some extent, how wide of a span the eggs were being laid over. We have a student who will be picking up this study as part of her broader dissertation work, and we hope to have some results (even if to say that it did not work) within the next two years.

Minor comments:

L77-79 "Further blurring... or heavily hunted":

Why is the use of other birds relevant to understanding the early dispersal of chickens? Please explain.

We thank the reviewer for pointing out an area where our text was not clear and we have attempted to fix it by adding a clarifying sentence.

L82-84 "In this manuscript ,... AD1220":

What do you mean by "success" here? Are the results of morphological analysis consistent with the results of peptide mass fingerprinting? Need an explanation in "Results" section.

Again, we agree with Reviewer 2 that our original wording here was not completely clear, and we have clarified it now.

L84-85 "The lack of ...archaeological sites":

Are there archaeological sites from the same era or earlier that attempted to collect eggshells with a sieve like this research? Please explain with adequate references. The "a rapid rise of egg-laying and chicken rearing" claim is irrelevant if there has been no such sieve sampling in earlier times.

The reviewer is again correct that our wording could have been clearer, and we have attempted to do so. Note that several of the authors have worked at other sites in Central Asia, most notably Spengler, who has worked on Bronze Age assemblages for much of his career. In addition, many other sites in the broader region have been published where flotation has been conducted and none of these sites have produced eggshells.

L137-138 "the bird is... AD 300 13)"

The Yayoi period is inadequate for Korea. Ref 13 would be inadequate here.

We thank the reviewer for catching this error; Reviewer 1 caught the same error. We are not sure how the citations were mixed here, but it has been rectified.

Figure 1: Please show the site names in the caption. For archaeological sites in Uzbekistan and Tajikistan, indicate them numerically on the map and indicate the site name in the caption.

Labeling all 50 sites in the main manuscript would take up considerable space in an article that is already over the word limits. We labeled all the site names for the sites that are relevant to the text and have added the site names with the numbers in the SI section. We appreciate the reviewer's concerns here, but we need to find ways to cut the word count down and none of those sites are directly relevant to the discussion in the manuscript. We hope the reviewer understands.

L174-176 "Additionally, ... evidence for absence)":
The description is not a result of this study.

Following the reviewer's comment above, we have clarified that all other sites that Spengler and Mir Makhamad have worked on in Central Asia, along with work by colleagues, such as Naomi Miller, George Willcox, or Reinder Neef – collectively representing many Bronze and Iron Age sites – have not resulted in the recovery of eggshells.

L179:
Are you studied samples from 13 sites?

Reviewer 3 has pointed this out as well, and we have chosen to change it to 13, but it really comes down to whether Paykend 1 and 2 are considered the same site or not.

L185-187:
In Paykend 1, the ubiquity is 30%. It is obviously less than 50%.

We again thank the reviewer for catching this error and we have fixed it, Paykend 2 and 3 both have 50% ubiquity and Bash Tepa in 70%. Much of the early layers at Paykend 1 are mudbrick fortification walls, so the overall assemblage from the early layer had lower densities of material.

Table 1:
Please explain what is the "Number of Samples".

We have added a clarifying point below the table.

L217-223:
Were the samples identified as Anseriformes considered to be other than chickens in morphological examination?

Yes, morphologically, we had considered them chickens prior to running ZooMS on them. We have noted this in the text.

L405-406 “Chicken express... environmental zones”:
References are required.

We have added a citation that gives general limitations to chicken rearing.

Reviewer #3 (Remarks to the Author):

- What are the noteworthy results?

The article addresses the cultivation of chicken in Central Asia. A major strength is the molecular techniques and eggshell analysis that is presented to reassess chicken domestication. The review provides extensive detail of sites, dates, evidence found, and commentary regarding the acceptability of the evidence and reasons why. This is a huge manuscript when the supplemental material is considered and will be useful to researchers for many years.

We are glad to see that the reviewer sees the merits of our paper and the scale of this study.

- Will the work be of significance to the field and related fields? How does it compare to the established literature? If the work is not original, please provide relevant references.

This article is the most extensive review I have ever seen regarding the topic across existing literature. The information presented is significant and contributes greatly to the topic in archaeology, poultry science, molecular biology, and veterinary sciences. The work is entirely original, although the preprint of the manuscript is available also as Spengler et al.

Spengler III, R. N., Peters, C., Richter, K. K., Mir Makhamad, B., Stark, S., Fernandes, R., et al. (2022). When did the chicken cross the road: archaeological and molecular evidence for ancient chickens in Central Asia. Research Square, 1340382/v1. doi:10.21203/rs.3.rs-1340382/v1.

We are aware the *Nature* posted a pre-print of this manuscript last year, then after a very long review process, four positive reviews, revisions, and a second round of reviews (apparently delayed by a reviewer), which were also mostly positive, the editor decided to reject the paper anyway. They claimed that they were rejecting it on the grounds that a genetics paper in PNAS, which came out in the course of our long review process, reduced the novelty of our manuscript. As the process of review in *Nature* took so long, the pre-print is now a year old, and we hope to have the paper properly published soon.

- Does the work support the conclusions and claims, or is additional evidence needed?

Yes. No additional evidence is warranted as extensive details are presented

- Are there any flaws in the data analysis, interpretation and conclusions?- Do these prohibit publication or require revision?

None that I can identify

- Is the methodology sound? Does the work meet the expected standards in your field?

Yes, the methodology regarding their approach to reviewing the sites and conducting the eggshell analysis is sound and explained in great detail exceeding expected standards.

- Is there enough detail provided in the methods for the work to be reproduced?

Absolutely. The information regarding research steps and the material and equipment used will allow the work to be reproduced.

General Comments

- **The research presented results from a huge endeavor by the authors. The manuscript could basically be divided into two articles even. The first about the eggshell analysis and second about their review of the literature and evidence**
- **In my view, the points that the authors make are often hidden in their extensive review in the Discussion. This is often unavoidable when reviewing so much information. Perhaps a review of these points could be made in the Conclusion**

Again, we are very happy to know that the reviewer sees so much merit in our work.

Introduction 62-151

- **78 The author may consider rewriting the sentence to use the term “husbandry” instead of “cultivated”**

We had provided a specific definition of cultivated, as we were using it to imply something different from husbandry. As this was unclear to the reviewer, we opted to not use either word and have now replaced cultivated with maintained, we hope this is suitable for the reviewer.

- **82 suggest keeping all information about the manuscript together. Move the sentence “In this manuscript....” to the beginning of 105**

We have moved the section of text down one paragraph.

- **89-104 suggest moving this paragraph and starting a new one after the sentence in 79-82**

Assuming we understand what the reviewer is thinking here, then we agree with her/him and have split the paragraph accordingly.

Dissemination across the Ancient World 128-151

- **142 suggest more direct and concise sentences such as “Debate of early chicken spread focus on the**

reliability...”

We are always happy to make sentences more direct and concise, so we accepted the reviewer’s exact wording suggestion.

- 149-150 not really sure if (Figure 1) is necessary as this could be referred to in an added reference

We have removed the call out to Figure 1, following the reviewer’s suggestion.

- 151 suggest adding an exemplary reference (“textual source”) at the end of the sentence

We have pulled up the citation for Pliny the Elder, which we also cite later in the text along with several other top Classical sources that discuss Chickens.

Results

Archaeological Eggshells 170-191

- 172 add “(Figure 1) after “in this study,”

We have moved the Figure 1 call out to here, following the reviewer’s suggestion.

- 175 for conciseness, phrases such as “bearing in mind” should be avoided. Use “considering that the...” There may also be other sentences in the manuscript that could be edited to remove/change similar phrases

We have changed the phrase as suggested by the reviewer.

- 179-180. I believe there are 13 sites in Table 1

Reviewer 2 pointed this out as well and we have changed it accordingly, it really just comes down to whether Paykend 1 and 2 are considered the same site or not.

Discussion 230-285

Evidence for Non-Seasonal Egg Laying 232-294

- 234 says “We present evidence for ancient chicken eggshells from 13 different archaeological settings, spanning a period of a millennium (Table 1).” In Table 1, 13 sites are listed.

We have changed the text to follow the reviewer's wording suggestions and we have also switched it to 13 sites, as noted above.

- 250-253 suggest rewriting as "The resulting evidence suggests that these chickens
- expressed shifts in reproduction from the wild and were producing eggs at a regular rate for a significant part of the year; although, these data do not allow us to specify the duration or abundance of laying."

We have changed the wording following the reviewer's suggestion, the previous wording was part of our attempt to remain cautious – specifically to defend against reviewers requesting more evidence to be absolutely certain before presenting a hypothesis – as Reviewer 2 has done.

Symbolic and Economic Prominence of the Chicken in Central Asia 296-223

- 298-310 This section could be improved by beginning with the reporting and interpretations by the authors rather than other studies to make it stronger. The specific results of the study and the support the evidence provides become hidden in the extensive review.

We have attempted to make some of these changes, but our goal in this SI section was to present the state of the field, as there really are no good current summaries. Our results are more fully presented in the main text.

- 335-337 Not sure what the need is for including the names of the zooarchaeologists who identified remains

We have removed the names, we just wanted to point out that two separate scholars had conducted a zooarchaeological study on material from the same site and both came to the conclusion that chickens were present.

Conclusions 387-428

- 406 perhaps say " The eggshell data we present from these sites..."

We have again accepted the reviewer's suggestions for changing of the words and are happy with the new wording.

References 431 566

- I leave the checking and accuracy to the copy editors reviewing the manuscript.

Methods 569-672

- Nice details are presented in the Methods to explain the procedures for peptide mass fingerprinting
- 584-586 Should the sentence “The shells were mostly white in color; although, some from Tashbulak appeared to have speckling, which may represent post depositional discoloration or possibly a distinct variety of chicken.” be placed in the Results section?

As Reviewer 1 requested a discussion of the color of the shells and because this sentence was buried in the methods, we have moved it into the main text. We thank both reviewers for the suggestion.

Methods References 675-759

- I leave the checking and accuracy to the copy editors reviewing the manuscript.

Figures and Tables

Figure 1. Should clarify the placement of points 45-49, and it is confusing between 48 and 49 and the cluster of 45-47 contains four points. This is always a problem in a distribution figure such as this.

We see the reviewer’s concern and have attempted to fix it in the new figure; we had not paid as much attention to these details, as these sites are not important to the overall discussion of the paper and at the level of zoom in the map, each dot covers a massive geographic area.

Figure 2 Pictures e and f could benefit from adding size bars similar to a and b

As e and f are scale zoom-ins on a and b, the size bar for a and b should be sufficient. If it is not sufficient, we will have to retake the photos as the size bars are auto-generated by the Joel SEM. We could retake the images, but we are not sure that really seems necessary.

Table 1: not sure how Nature Communications handles empty table cells

Supplemental Material

Extended Data Table 1: Key Archaeological Sites Supporting the Westward Spread, Linked to Figure 1.

- Listing and information are fine

Extended Data Table 2: Sample numbers, site and context information, and final identification of the eggshell fragments included in this study.

- Listing and information are fine

Extended Data Table 3: Proteins identified in sample CP565 (Afrasiab) with a LogProb ≥ 5 , and at least 2 unique peptides.

- Listing and information are fine

Supplementary Discussion 1: The Chicken Domestication Debate

- This section is an amazing review of the status of chicken domestication. But with all the information presented, the stances of the authors are not easily found or discerned. Statements at the beginning of each section or paragraph would make the authors' interpretation and stance stand out.

We thank the reviewer for recognizing the scale of our review here. We had tried to be as impartial as possible in this section, as it is supposed to be a review of the existing data, whereas our views are presented in the main text. However, we have read it all over again to verify the clarity and to add a few sentences to show our interpretations.

- May be useful to provide the names of other researchers in the first section similar to your references of Best, Peters, West & Zhou, and others in the later paragraphs. It is nice to be able to keep track of the researchers and their debates without looking up the reference number

This is a common issue with these numbered citations, as our early versions of the text all had in-text citations and it was easy to follow, but after changing the names to numbers, much of this clarity is lost. We have attempted to reach a middle ground here and hope that the reviewer is happy with the current version.

Widely Conflicting Claims

- Which claim is most acceptable to the authors?

The goal in this section was to show how different all the conclusions drawn by these researchers have been. The goal was more to show the state of the field, as opposed to providing an argument here. Although, if pushed to make a stance, we would all favor the most recent conclusions published by Peters et al. 2022, which are really quite different from any other previous conclusions.

Dispute over the Chinese Center of Origin

- What evidence and date are the most acceptable to the authors?

In this section, we actually were extremely clear in our rejection of the early Chinese Center of Origin argument – but, of course, lots of scholars have rejected it before us. There is just too much evidence now against it.

Lack of Resolution

- What dispersion route is most acceptable to the authors?

We have added a sentence that clearly states our view here. We have a similar sentence in the main text of the manuscript, so the reader cannot fail to understand where we stand.

Supplementary Discussion 2: Dissemination across the Ancient World

- Can the authors state which study they consider to provide the best evidence of dissemination?

Supplementary Discussion 3: The Sites Discussed in this Manuscript

- Suggest revising the title of this section to: “Supplementary Discussion 3: Sites containing Eggshell Discussed in this Manuscript”

We again agree that the reviewer’s wording is better than what we had and changed the text accordingly.

- Suggest providing the sites in the order they are presented in Table 1

We have rearranged the sections as recommended by the reviewer.

- Ming Tapa is not found in Table 1

Well... This is a serious error. I have no idea how a full site is missing on our final data table, but it is the last site we added to the assemblage. We thank the reviewer for catching this error.

Supplementary Discussion 4: Data and Results of Peptide Mass Fingerprinting of the Eggshells

- Is this the take-home message for this section: “The closest relative to chicken present in Uzbekistan, ring-necked pheasant, can be distinguished from chicken using published markers, and we find only evidence corresponding to chicken specific peptides in the LC-MS/MS, so, even without having references from local wild taxa, we can securely use these MALDI peptide markers to identify a sample as chicken as opposed to local, wild taxa.” Although it is a matter of presentation style in scientific writing, the authors may consider placing their definitive statements at the beginning of a section rather than the end. I only recommend this, because in my recent two article submissions this point was raised to improve the manuscript.

References

- 92 Should it be “Brown”?

It is Brown, we checked again and it is written correctly.

- Ref 117 should be Eda not Edu

Yes, the reviewer caught another error and we are grateful.

We would like to say an extra thank you to Reviewer 3; you have clearly spent an immense amount of time reading our manuscript in detail and you have provided very helpful comments and revisions.

Reviewers' Comments:

Reviewer #1:

Remarks to the Author:

The authors have made revisions to address my concerns. Especially, the updated supplementary files are informative for researches. It is expected that, the future investigation of ancient DNA from those sites in southern Central Asia can depict more detailed pictures about the dispersal and breeding history of chickens. Herein, I recommend the manuscript for a potential publication of Nature Communications.

Reviewer #2:

Remarks to the Author:

I read through the revised paper. I agree with some of their revisions, but there are some parts I disagree with. In Supplementary Discussion 1, authors said as follows: "Peters et al. choose to remain highly conservative in their judgement of what data points to accept. They state this choice clearly, and emphasize that non-directly dated specimens, and specimens without clear morphological or genetic criteria for identification should be taken with caution or rejected. Peng et al. criticize them for doing this. We side with Peters et al., and suggest that all archaeological investigations should follow their criteria". However, in my opinion, the authors attitude towards eggshell identification and the "non-seasonal laying" debate is far from conservative. Details are given below.

Major comments:

1. Lack of reliable identification for eggshell:

Authors conducted morphological and peptide mass fingerprinting identification in the manuscript. However, results of morphological identification are not fully explained. They described the size and morphology of the archaeological eggshell breathing pores as consistent with modern chicken eggs (L581-584), but that description is not sufficient for those eggshells to identify chicken eggshells. The number of morphologically identified samples is unknown (this is also a problem), while only 16 eggshells from 4 sites were identified as chicken by peptide mass fingerprinting. Other archaeological eggshell fragments may also belong to other species, as two of the eggshells were identified as 'Anseriformes' by peptide mass fingerprinting. Therefore, it is inappropriate to argue that the results presented the paper indicate that chicken eggshells were found at 12 sites. On the other hand, the samples subjected to peptide mass fingerprinting also do not appear to provide sufficient evidence to distinguish them from chicken (*Gallus gallus*) eggshells. Although LC-MS/MS is applied to one eggshell sample, it is logically impossible to identify chickens rather than other local Phasianidae birds without data on local Phasianidae species eggshell proteins. Ring-necked pheasants are the most closely related to chickens in the local Phasianidae species, but this does not guarantee that chickens and ring-necked pheasants are the most closely related for all peptide fragments. Furthermore, even if pheasants and chickens can be distinguished according to PMF markers found in previous studies, can chickens be distinguished from other *Gallus* junglefowl? To identify it as *Gallus gallus*, it must be shown that it is not another *Gallus* junglefowl. Revision is required.

Author's reply:

We agree with the reviewer that it is important to be able to uniquely identify the eggshell remains. In an ideal world we would have reference material for every locally present species in the Galliformes family, but this was not possible. Therefore, we created a database that includes all available protein data from all seven Phasianidae species (subfamilies *Perdicinae* & *Phasianinae*) that are locally present in Uzbekistan (see Supplementary Discussion 4).

We know from previously published data that there are differences between the MALDI spectra for chicken, pheasant, quail and other species in the Galliformes Order (Presslee et al. 2017). For the LC-MS/MS searching we knew already from the MALDI data that the sample belonged to a galliform. Therefore, we were focused on including galliform species. We included some additional species that were closely related to species locally present based upon data from avibase that had high quality

data available, unfortunately many of the most obvious species do not have published proteomes. In order to combat that the database also includes key eggshell proteins that are available from all species. This single analysis represents all eggshells identified as chicken, and they all have the same set of peptide markers identified from their MALDI spectra which correspond to previously published chicken markers also verified by LC-MS/MS.

Finally, we have checked all peptides for uniqueness to chicken using blast against the larger NCBI database. We are confident that the peptides we are seeing in high abundance in the MALDI derive uniquely from chicken and not from other locally present wild galliform species that lacked published MALDI markers, which was the largest concern given the published MALDI markers that had already clearly identified the sample as *Gallus gallus*. We cannot exclude that the eggshell does not derive from *Gallus junglefowl*. However, the geographic range of this species falls far out of the current region of study. Additionally, the number of eggshell fragments recovered from the sites makes it highly unlikely these eggshells were imported.

To address the reviewer's concerns, we have added these last three sentences to Supplementary Discussion 4.

Presslee, S. et al. The identification of archaeological eggshell using peptide markers. *Sci. Technol. Archaeol. Res.* 3, 89-99, doi:10.1080/20548923.1424300 (2017).

We have attempted to make it clear throughout the text that we do not believe that the morphological criteria for identification is not, in and of itself, enough to confidently say that these are chicken eggshells. However, we cite several studies that have used these morphological criteria to differentiate between chickens and other birds in the past. We are seeking a more reliable approach in this manuscript and we are rather confident that our current approach, which utilizes ZooMS (a highly reliable methods), descriptive morphology, art historical sources, historical evidence, and to a lesser extent zooarchaeology (note that proper zooarchaeological studies are underway at several of these sites and they are also finding clear evidence for chickens), achieves a reliable conclusion. We are very happy to accept that there may be non-chicken eggshells (possibly from domesticated geese) mixed into the assemblages, but it is beyond the scope of our study to analyze all of these eggshells. Follow-up aDNA studies are planned with a team under Greger Larson's direction, so over the next several years there will be additional clarity. Given that it has taken nearly a decade to collect this data and more than four years to process the eggshells and compile the historical and art historical sources, we hope that this reviewer will find our manuscript worth publishing.

Reviewer's reply:

I believe that eggshell protein data of other seven indigenous Phasianidae species are indispensable for identification of chicken eggshells. Given the other papers published in *Nature communications*, I do not think that asking the authors to analyze eggshell proteins from these Phasianidae is too high a demand. On the other hand, I admit that authors are taking the best possible approach in the current absence of data for these species. I'll leave it up to the editor to decide whether or not chicken eggshell identification by ZooMS is adequate in the journal.

However, I believe there is absolutely no evidence that the eggshells found from the other eight sites (not identified by ZooMS) belong to chickens. As is clear from the authors' responses to my comments, the authors do not believe in the morphological identification of eggshells at all. How is the author able to identify chicken eggshells without using the ZooMS approach? In addition, some of the eggshells that the authors considered to be from chicken by morphological analysis were identified as Anseriformes by ZooMS. Although it is said that SEM observation was not performed on the identified Anseriformes eggshell, then, how the authors identified the eggshells? Please clarify how many eggshells were analyzed by morphological approach with and without SEM. I also pointed this out in my previous comment. Revision is required.

2. Lack of reliable dating for eggshell:

As reviewed by the authors, reliable identification and direct dating of chicken bones is essential to uncovering the presence of chickens at the site at that time. Is there evidence that the eggshells analyzed in this study were not contaminated from a later period? In addition, the inconsistency in

sample ages undermines the reliability of the data. For example, the date of samples from Bash Tepa is ca 3rd century BC- 1st century AD in the text (L112-113) but 200-1 BC in Table 1 and 400-0 BC in Extended Data Table 1. Revision is required.

Author's reply:

We appreciate the reviewer's concerns regarding the dating, and we have paid for additional rushed radiocarbon dates. These new dates clearly illustrate the oldest ages of our chickens – we added the dates to the paper along with clarification of the discussion. We had not run radiocarbon dates before, as our broader project has run more than two dozen dates on seeds from these sites and the chronologies are fixed with many more radiocarbon dates run by the project excavators, along with a very well-worked ceramic and numismatic seriation. Bash Tepa, in particular, has a highly detailed chronology sorted out through four-years of excavation, including an additional nearly two dozen dates, run by Soren Stark and his team. We absolutely do not dispute the possibility that some of the eggshell fragments have been turbated – we have enough experience with unexpected archaeobotanical material being stratigraphically out of place in sites – however, when dealing with thousands of eggshells from dozens of contexts at 13 different sites, the chances that everything is out of place is extraordinarily improbable.

We hope the reviewer finds the new dates suitable to put all concerns at ease, and again, we thank the reviewer for raising such concerns.

Reviewer's reply:

It's great that the eggshell has been dated. Please clarify whether the dated eggshell was identified as a chicken. Dating an unknown species eggshell is not so meaningful in this case. The ages of eggshells from Paykend 1 are significantly older than those indicated in Table 1. If the chronology of the site has changed, please revise Table 1.

3. Lack of the evidence for non-seasonal egg laying:

The idea is very interesting. However, the evidence is, in my sense, very weak. The only evidence shown by authors is "artefacts recovered in many different archaeological contexts are more likely to represent frequent-occurrence events, as opposed to seasonal or semi-annual events" (L240-242). Judging from the references given there, only two closely related authors mention this pattern in their books. Is this a well-accepted proof for archaeobotanists? Also, how much ubiquity does it take to say that it is not a seasonal or semi-annual event? In Paykend 1, the ubiquity is 30%. Is this enough to say frequent-occurrence events? I think there is a big gap between frequent-occurrence events and non-seasonal egg laying, and common food over a greater period of the season (L247-248).

Author's reply:

We recognize that this is a logic argument, and we have nothing more to add to it at this point. If the reviewer is unwilling to accept our argument, we simply cannot provide more proof at present. However, we strongly believe that this reasoning is well-accepted in the field of archaeobotany. Additionally, we simply cannot imagine an alternative option, we would go so far as to say that we find it highly improbable that our reasoning is wrong. Despite our confidence, we have attempted to remain as cautious as possible in the wording and have tried to avoid specifics, such as how many eggs a year a chicken might have been laying at this time or how long the laying season might have been. That said, a wild laying state would suggest that one brood a year, or roughly six chickens, was the norm; finding thousands of shell fragments across dozens of different anthropogenic contexts seems unlikely given the wild laying state. Further backing up are discussion, are historical sources from only a few centuries later in time that clearly discuss non-seasonal egg laying patterns among chickens in the Classical world. We appreciate the reviewer's concerns here, but we hope that they will be lenient on this point, as we do think it is important to leave it in the manuscript, as some discussion of the cultural aspects of this study is important to us.

As a side note, we are in the process of brainstorming a check for this theory that involves isotopes from eggshells and banded-sampling of isotopes on ungulate teeth from the same context. While there

will be many complicating factors in such a study, it may show, to some extent, how wide of a span the eggs were being laid over. We have a student who will be picking up this study as part of her broader dissertation work, and we hope to have some results (even if to say that it did not work) within the next two years.

Reviewer's reply:

There were no revisions from the authors on this point. Authors didn't answer my questions "how much ubiquity does it take to say that it is not a seasonal or semi-annual event?" and "In Paykend 1, the ubiquity is 30%. Is this enough to say frequent-occurrence events?"

Grains were supposed to be grown for the purpose of eating them, but eggshells were not edible. Can grains and eggshells be considered the same? Although the authors said that "we have attempted to remain as cautious as possible in the wording and have tried to avoid specifics, such as how many eggs a year a chicken might have been laying at this time or how long the laying season might have been", this is normal. If author's have information about the number of eggs per year and the laying period, the authors only need to state it clearly.

It is unclear when the authors assume non-seasonal egg laying started. Additional dating has shown that the oldest (chicken?) eggshell dated from 515-392 BC. Do the authors presume that chickens were doing non-seasonal egg laying from this period? According to the authors' review, however, few chicken bones have been found in Kyzyl Tepa from the 6th to 4th centuries BC. And that is the only record in this period. Do the authors think that although the egg-laying season of chickens has become longer, chickens have not been used much?

Minor comments:

L77-79 "Further blurring... or heavily hunted":

Why is the use of other birds relevant to understanding the early dispersal of chickens? Please explain.

Author's reply:

We thank the reviewer for pointing out an area where our text was not clear and we have attempted to fix it by adding a clarifying sentence.

Reviewer's reply:

Misidentification of chickens with other Phasianidae species is evident in author's review. On the other hand, are there any examples where ducks or geese were misidentified as chickens? And, is it common? Please introduce by citing appropriate papers. The size of goose and early stage of domestic chickens vary considerably.

L82-84 "In this manuscript ,... AD1220":

What do you mean by "success" here? Are the results of morphological analysis consistent with the results of peptide mass fingerprinting? Need an explanation in "Results" section.

Author's reply:

Again, we agree with Reviewer 2 that our original wording here was not completely clear, and we have clarified it now.

Reviewer's reply:

How do the authors combine morphological data with PMF? Please explain it in the results.

L84-85 "The lack of ...archaeological sites":

Are there archaeological sites from the same era or earlier that attempted to collect eggshells with a sieve like this research? Please explain with adequate references. The "a rapid rise of egg-laying and chicken rearing" claim is irrelevant if there has been no such sieve sampling in earlier times.

Author's reply:

The reviewer is again correct that our wording could have been clearer, and we have attempted to do so. Note that several of the authors have worked at other sites in Central Asia, most notably Spengler, who has worked on Bronze Age assemblages for much of his career. In addition, many other sites in the broader region have been published where flotation has been conducted and none of these sites have produced eggshells.

Reviewer's reply:

It's great to have so much flotation data already. However, I and the readers do not know which ruins correspond even if it is written as "(Spengler has done extensive flotation and water screening work on more than ten earlier sites in Central Asia over the course of his career)" and "and many of our authors have spent years working on earlier sites in Central Asia". Add the data to Table 1 with appropriate citations. Negative data that no eggshells were found after flotation are important to say "a rapid rise of egg-laying and chicken rearing"

L137-138 "the bird is... AD 300 13)"

The Yayoi period is inadequate for Korea. Ref 13 would be inadequate here.

Author's reply:

We thank the reviewer for catching this error; Reviewer 1 caught the same error. We are not sure how the citations were mixed here, but it has been rectified.

Reviewer's reply:

As far as I know, Peters et al (2022) did not mention archaeological sites in Korea. Again, the Yayoi period is inadequate for Korea.

Table 1:

Please explain what is the "Number of Samples".

Author's reply:

We have added a clarifying point below the table.

Reviewer's reply:

Thank you for your clarification for this point. Then, how many eggshell samples do you observe by morphological approach? Please show the "Number of eggshell fragments" for LVD-HA-K7, Kafir Kala, and Ming Tapa.

L405-406 "Chicken express... environmental zones":

References are required.

Author's reply:

We have added a citation that gives general limitations to chicken rearing.

Reviewer's reply:

I cannot find out the citation. Please check it again.

Reviewer #3:

None

We start by addressing the comments that reviewer 1 made in regards to the comments of reviewer 2, as these are insightful and discreet. We hope that reviewer 1 feels that we have addressed all of his/her concerns.

Reviewer One

Reviewer's Comment: The authors have made revisions to address my concerns. Especially, the updated supplementary files are informative for researches. It is expected that, the future investigation of ancient DNA from those sites in southern Central Asia can depict more detailed pictures about the dispersal and breeding history of chickens. Herein, I recommend the manuscript for a potential publication of Nature Communications.

The genomic data for other Gallus junglefowl species are available (Wang MS. et al 2020. Cell Res; PMID: 32050971; PMID: 33346111). The authors may screen the nonsynonymous mutations in the genes investigated by LC-MS/MS in other Gallus junglefowl species. The results are helpful to illustrate that chickens can be distinguished from other Gallus junglefowl or not.

We are aware of the Wang et al (2020) manuscript and we have cited it; we are also aware of these keyed out mutations. Although, it is a bit beyond the expertise of anyone on our team to judge the reliability of these markers. The corresponding author has already been in contact with Gregor Larsen and has promised to send a sampling of our materials to him after the publication of this manuscript. The manuscript has been in the review process for a year and a half, but once it is published we will send our samples to their chicken aDNA project to integrate with their ongoing studies. Currently, Central Asia is absent in their studies, and they have very few data points from West Asia more broadly, illustrating the importance of our manuscript.

Regarding the possible misidentification of our samples as red junglefowl, what this scenario would suggest is: that someone or several people captured a tropical South Asian bird and carried it four to six thousand kilometers to the north where they raised it in a hyper arid desert. They then raised it to a high enough population level to leave high ubiquities of eggshell fragments in the anthropogenic sediments around their sites. They did all of this while also raising chickens, as we know from historical and art historical sources (neither of which describe or depict anything like a wild jungle fowl). Ultimately, this scenario is so improbable that even if our eggshells had the indicator alleles for domestication, we would question the reliability of the alleles as indicators before we would believe that these eggshells are from wild junglefowl. While we do believe that follow-up research is needed, we worry that some of the scenarios that reviewer 2 is asking us to test are too improbable to merit consideration – especially, given the difficulty or impossibility in testing them.

Reviewer's comment: Although the authors tried to explain details in Supplementary Discussion 4, it is better to declare their efforts in the text or supplementary files. For instance, to add the results of non-chicken (e.g., ring-necked pheasants) peptides generated in this study in Figure 4. It is also advisable to describe the ZooMS results for each of eggshell samples in a new supplementary table. The raw data can be deposited into open database (e.g., <https://www.scidb.cn/en>).

We agree with the reviewer and have tried our best to address the raised points. We hope that this will alleviate any remaining concerns. We have rerun all the data, as presented in the new SI table. Notably, we have added more information to the main text, and added a new SI table Raw MALDI data was uploaded to Zenodo, and MS/MS data has been uploaded to ProteomExchange. Both will be made publicly available upon publications of the manuscript.

Unfortunately, we were not able to add the spectrum of a ring-necked pheasant or any other non-chicken Galliform to Figure 4. Raw spectra files from previous studies were not made available with those publications, and we do not have a sample identified as non-chicken Galliform in our own sample set to use for this purpose. Therefore, we do not have a proper spectrum that we can use to add to the figure; we do have many specifically noted indicator proteins, but not a full published spectrum.

Reviewer's Comment: It is not clear for the dating results in this manuscript. Is it possible to date some eggshells? If not, why? Some explanations can included in Supplementary Discussion. To address the concerns, I also suggest the authors to providing the dating results generated in this study in a new supplementary table.

It is, of course, possible and we have now dated the oldest eggshells in the assemblage. We have, as reviewer 2 also requested, taken these dates from our SI section and added them at the opening of the main text. The chronologies for all of these sites have been constructed in unison with the excavation teams. Spengler's broader team has run more than 30 radiocarbon dates over the past few years in an effort to fill in any remaining gaps in these chronologies. Several dozen more radiocarbon dates have been run by the excavation directors themselves. Additionally, most of these sites contain ceramics that fit into very clear and well-tested chronologies and all of the later sites have coins that provide even more precise dates than can be obtained from radiocarbon dating. One of the authors here is also working on a separate paper that will pull together all published archaeological radiocarbon dates for Central Asia, further clarifying these chronologies.

Reviewer's Comment: It is really a challenge to characterize the non-seasonal egg laying. Basically, the non-seasonal egg laying could increase the number of eggs. Is it possible to infer or evaluate, largely, the number of eggs from the data of eggshells? For instance, the authors can evaluate the sizes for eggshell fragments and the contents. If grounding modern eggshells into such sizes and then mixed into soil, is it possible to estimate the numbers of eggs?

We do not believe that there is any feasible way to quantify this interpretation. Also, given that reviewer 2 is unwilling to believe the interpretation as a general statement, making it more specific should make it more falsifiable and, therefore, easier for the reviewer to attack – hence, all we can do is try to explain our interpretation of the data as a logic statement, as we have attempted to reiterate in our response to reviewer 2. We realize that we may fail to convince the reviewers of this, but we have thought extensively about this question and we have not come up with any clear way to test it. See – response to reviewer 2 below.

Reviewer Two

Reviewer's Comment: I read through the revised paper. I agree with some of their revisions, but there are some parts I disagree with. In Supplementary Discussion 1, authors said as follows: "Peters et al. choose to remain highly conservative in their judgement of what data points to accept. They state this choice clearly, and emphasize that non-directly dated specimens, and specimens without clear morphological or genetic criteria for identification should be taken with caution or rejected. Peng et al. criticize them for doing this. We side with Peters et al., and suggest that all archaeological investigations should follow

their criteria". However, in my opinion, the authors attitude towards eggshell identification and the "non-seasonal laying" debate is far from conservative. Details are given below.

We are sorry to hear that this reviewer is still upset with our research and interpretations – we worked hard to address all of her/his points during the last round of edits.

After careful consideration and a re-reading of the Peters et al. paper, we have removed the comment that the reviewer points out here – we recognize that Peters et al. really were not conservative in their interpretations of the data. We remain hesitant to add a critique of their work here, as they include a number of powerful academics, and we do not wish to upset them. We feel that our interpretations are relatively strong in comparison.

Regarding the seasonality of egg laying, we recognize that there is nothing more we can do to try to appeal to this reviewer. It is based on a logical assessment of the data and not a quantifiable analysis. The reviewer asks us to quantify this interpretation, but it is simply not a quantifiable statement given either the data that we have at hand or any data that we could conceive of producing in the future. We do specifically state in the main text: "these data do not allow us to specify the duration or abundance of laying". Hence, we stand by our interpretation of the data, and hope that the reviewers will choose to allow us to publish, in which case the reviewer is welcome to write a critical response to our paper. Disagreeing with an interpretation in science is acceptable, as long as we all agree on the data.

Reviewer's Comment: I believe that eggshell protein data of other seven indigenous Phasianidae species are indispensable for identification of chicken eggshells. Given the other papers published in Nature communications, I do not think that asking the authors to analyze eggshell proteins from these Phasianidae is too high a demand. On the other hand, I admit that authors are taking the best possible approach in the current absence of data for these species. I'll leave it up to the editor to decide whether or not chicken eggshell identification by ZooMS is adequate in the journal.

Following reviewer 2's response to this, we have added another table that hopefully makes it easier to understand that we have a very clear identification and that no other wild relative is even remotely probable as a misidentification in this case. We have also rerun all the data and presented the new data in a new SI table, the new data also illustrates the same conclusions. As we note previously, protein sequences from three of the common wild members of the pheasant family are available for download from UniProt and NCBI – the chukar, the ring-neck pheasant, and the desert partridge. The remaining four species are rare and it would be impossible to try to track down eggshells from the wild (not to mention unethical), furthermore, their habits and distributions make them highly improbably as targets for frequent nest raiding by humans. Beyond that, all of these four species are partridge relatives (quail or partridge/grouse) and there are peptide sequences available from UniProt and NCBI for multiple taxa that are significantly more closely related to them than they are to the chicken; hence, we can nearly unequivocally rule all of them out.

However, I believe there is absolutely no evidence that the eggshells found from the other eight sites (not identified by ZooMS) belong to chickens. As is clear from the authors' responses to my comments, the authors do not believe in the morphological identification of eggshells at all. How is the author able to identify chicken eggshells without using the ZooMS approach? In addition, some of the eggshells that the authors considered to be from chicken by morphological analysis were identified as Anseriformes by ZooMS. Although it is said that SEM observation was not performed on the identified Anseriformes eggshell, then, how the authors identified the eggshells? Please clarify how many eggshells were

analyzed by morphological approach with and without SEM. I also pointed this out in my previous comment. Revision is required.

SEM is not necessary for morphological identification, and in either case, we do not claim to have identified all of the eggshells in the assemblage – just a representative portion.

Reviewer's Comment: It's great that the eggshell has been dated. Please clarify whether the dated eggshell was identified as a chicken. Dating an unknown species eggshell is not so meaningful in this case. The ages of eggshells from Paykend 1 are significantly older than those indicated in Table 1. If the chronology of the site has changed, please revise Table 1.

As radiocarbon dating and MS analyses are destructive methods, we have dated a two shell fragments from two contexts where two other shell fragments were analyzed using ZooMS. They are indeed a century or two older than we would have expected, but after more than a decade of excavation work at the site, the excavators are just now reaching contexts that they suspect are older than the overlaying medieval urban site.

Reviewer's Comment: It's great to have so much flotation data already. However, I and the readers do not know which ruins correspond even if it is written as "(Spengler has done extensive flotation and water screening work on more than ten earlier sites in Central Asia over the course of his career)" and "and many of our authors have spent years working on earlier sites in Central Asia". Add the data to Table 1 with appropriate citations. Negative data that no eggshells were found after flotation are important to say "a rapid rise of egg-laying and chicken rearing"

As this manuscript has already been expanded to the length of several normal publications, we feel that the request of synthesizing all previous research in archaeobotany is beyond the scope of our study. That said, several recent publications have already synthesized portions of this work, and a database to synthesize it all is underway by one member of our team. We have cited two of these publications, each has a table that should provide the reviewer with what he/she requests.